# *Candida albicans* gains azole resistance by altering sphingolipid composition

Jiaxin Gao[1,2,3], Haitao Wang[4,5], Zeyao Li[1,6], Ada Hang-Heng Wong [4], Yi-Zheng Wang[1], Yahui Guo[7], Xin Lin[7], Guisheng Zeng[5], Haoping Liu[3], Yue Wang [5,8] & Jianbin Wang [1,2]

Fungal infections by drug-resistant *Candida albicans* pose a global public health threat. However, the pathogen's diploid genome greatly hinders genome-wide investigations of resistance mechanisms. Here, we develop an efficient *piggyBac* transposon-mediated mutagenesis system using stable haploid *C. albicans* to conduct genome-wide genetic screens. We find that null mutants in either gene *FEN1* or *FEN12* (encoding enzymes for the synthesis of very-long-chain fatty acids as precursors of sphingolipids) exhibit resistance to fluconazole, a first-line antifungal drug. Mass-spectrometry analyses demonstrate changes in cellular sphingolipid composition in both mutants, including substantially increased levels of several mannosylinositolphosphoceramides with shorter fatty-acid chains. Treatment with fluconazole induces similar changes in wild-type cells, suggesting a natural response mechanism. Furthermore, the resistance relies on a robust upregulation of sphingolipid biosynthesis genes. Our results shed light into the mechanisms underlying azole resistance, and the new transposon-mediated mutagenesis system should facilitate future genome-wide studies of *C. albicans*.

[1] School of Life Sciences, Tsinghua University, Beijing 100084, China. [2] Centre for Life Sciences, Tsinghua University, Beijing 100084, China. [3] Department of Biological Chemistry, University of California, Irvine, CA 92697, USA. [4] Faculty of Health Sciences, University of Macau, Macau, China. [5] Institute of Molecular and Cell Biology, Agency for Science, Technology and Research, Singapore 138673, Singapore. [6] Peking-Tsinghua-NIBS Joint Graduate Program, Tsinghua University, Beijing 100084, China. [7] Institute for Immunology, School of Medicine, Tsinghua University, Beijing 100084, China. [8] Department of Biochemistry, Yong Loo Lin School of Medicine, National University of Singapore, Singapore 117596, Singapore. Correspondence and requests for materials should be addressed to Y.W. (email: mcbwangy@imcb.a-star.edu.sg) or to J.W. (email: jianbinwang@tsinghua.edu.cn)

Candida albicans is a major fungal pathogen of humans, affecting millions of people and kills hundreds of thousands worldwide annually[1]. Mortality rates of invasive C. albicans infection remain high despite the treatment with existing antifungal therapies. Multidrug-resistant Candida species and strains are rapidly emerging and spreading globally, threatening to render our entire antifungal arsenal obsolete[2]. Azoles are common first-line drugs against most Candida species in systematic infections, which act by inhibiting the 14α-demethylase Erg11 in the ergosterol biosynthesis pathway and causing the accumulation of the toxic sterol 14,24-dimethylcholesta-8,24(28)-dien-3β,6α-diol (DMCDD) that permeabilizes the plasma membrane[3]. Extensive research has demonstrated that fungal pathogens can quickly evolve various resistance strategies after exposure to an antifungal agent. C. albicans can gain resistance to azoles via genetic alteration of the drug target Erg11, upregulation of efflux pumps Cdr1, Cdr2, and Mdr1, and inactivation of Erg3 that synthesizes the toxic sterol[4,5]. However, our mechanistic understanding of antifungal resistance is incomplete[2]. It remains unclear why some fungal species are intrinsically resistant or easily acquire resistance to multiple antifungal drugs. Thus, it is imperative to identify and understand the molecular mechanisms that govern antifungal resistance.

Genome-wide genetic screening is a powerful tool to identify genes involved in a biological process. However, it has been difficult to apply these technologies to C. albicans due to its diploid genome. Although several recent studies conducting large-scale genetic screens of C. albicans mutant libraries resulted in important findings[6–10], investigation of many aspects of C. albicans biology still relies on inference from data obtained in the studies of Saccharomyces cerevisia[11,12]. This approach inevitably misses out genes that perform specific functions and determine unique traits in C. albicans. For example, S. cerevisiae is non-pathogenic and does not grow true hyphae which are the predominant morphology of C. albicans in host tissues and a key virulence factor[11]. Currently, most C. albicans genes with limited homology to S. cerevisiae remains uncharacterized[13].

Recently, some haploid C. albicans strains were isolated[14]. Although avirulent, the haploids possess most traits that define the species including the ability to undergo the yeast–hyphae transition, the white-opaque phenotypic switch, the formation of chlamydospores, and mating[14]. The haploids have opened up opportunities for scientists to explore large-scale, genome-wide genetic screening strategies in this pathogen. The recent application of CRISPR-cas9 technology in haploid C. albicans in the identification of new virulence regulators and genetic networks demonstrates the massive potential of using haploid C. albicans in genetic screens[15].

Transposon-mediated mutagenesis is a widely used genetic tool. Transposons are movable genetic units and have two essential components, a transposase and a pair of transposon-specific inverted terminal repeats (ITRs). One example is the piggyBac (PB) transposable element, first isolated from the moth Trichoplusia ni and later adapted to other organisms from bacteria to mammals[16,17]. PB transposase recognizes the ITRs and moves the entire content from one chromosomal locus and integrates it at a TTAA motif in a new locus[18]. PB transposition occurs via a precise cut-and-paste mechanism which leaves no footprints behind, resulting in a full restoration of the donor site[19]. Thus, a phenotype caused by a PB insertion can be easily verified by phenotype reversion. Furthermore, PB transposon inserts randomly throughout the genome[20].

In this study, we develop a highly efficient, PB-based mutagenesis system using stable haploid C. albicans strains. We apply this technology to conduct genome-wide genetic screens for genes involved in resistance to the antifungal drug fluconazole and discover a new resistance mechanism.

## Results

**Construction of the *piggyBac* transposition system in *C. albicans*.** We first optimized the *piggyBac* transposase (PBase) gene for use in C. albicans. We replaced all CUG codons with UUG because CUG encodes serine instead of the conventional leucine in C. albicans, added a 3×SV40 nuclear localization signal to its N-terminus, and controlled the expression using the tetracycline-inducible (Tet-On) promoter (Fig. 1a). Also, we mutated seven amino-acid residues to generate a hyperactive transposase according to previous studies[21]. We then integrated this CaPBase gene, using SAT1 as the selectable marker, at the ADH1 locus of the stable, uridine auxotrophic (Ura-) haploid C. albicans strain GZY803[22] to generate the YW01 CaPBase-expressing strain.

To test whether PB transposition would occur in C. albicans, we transformed the donor plasmid pPB[URA3] carrying the ITR-URA3-ITR cassette into YW01 cells that had been grown in the absence or presence of doxycycline (Dox) for 48 h and then spread the cells onto uridine-deficient (-Uri) plates. The transformation of Dox-induced cells led to the growth of numerous colonies, while the uninduced cells produced only a few likely due to the basal expression of CaPBase (Fig. 1b), an event that happened in <0.001% of the cells. This result indicated that the Dox-induced CaPBase excised the PB cassette from the plasmid and integrated it into the genome.

Previous studies in other organisms showed that PB excision did not leave any footprint behind[19]. To confirm this characteristic in C. albicans, we constructed a C. albicans strain in which we inserted the PB[URA3] cassette at a TTAA site (n.t. 709–712) within the open reading frame (ORF) of ARG4 (Fig. 1c), yielding YW02 that is arginine auxotrophic (Arg−) and uridine prototrophic (Ura+) and also expresses CaPBase from the Tet-On promoter at the ADH1 locus. Precise excision of PB would convert YW02 cells from Arg− to Arg+. After Dox induction for 48 h, we spread the YW02 cells onto glucose minimal medium (GMM) plates, picked 20 independent colonies, and confirmed that all 20 clones grew well on -Arg-Uri plates (Fig. 1d), indicating that the cells had excised PB[URA3] out of ARG4 precisely. This was confirmed by site-specific PCR using two pairs of primers as shown in Fig. 1c (Fig. 1e shows examples of 5 clones) and DNA sequencing analysis of the PCR products. To map PB insertion sites, we performed inverse PCR as described in Fig. 1f which allowed amplification of the DNA fragment encompassing a transposon junction. DNA sequencing analysis of the PCR products and homology search of the Candida Genome Database (CGD) located the PB insertion sites on all eight chromosomes (Supplementary Table 1), seven within ORFs and 13 in intergenic regions (Fig. 1g). Our results demonstrated random PB transposon insertion and precise excision in C. albicans.

**Ploidy and *PB* transposition efficiency analysis.** Auto-diploidization observed in C. albicans haploids has been a major concern regarding their applicability in genetic screens[14]. To evaluate ploidy stability during transposition, we first performed qPCR analysis to measure CaPBase expression levels at timed intervals in YW02 cells grown on solid media with or without Dox. The results demonstrated a sharp increase of CaPBase expression upon Dox addition (Fig. 2a). In parallel, we examined cell ploidy by flow cytometry and found that the cells retained 1N DNA content for at least 72 h (Fig. 2b) regardless of Dox addition, indicating a stable ploidy during the robust expression of

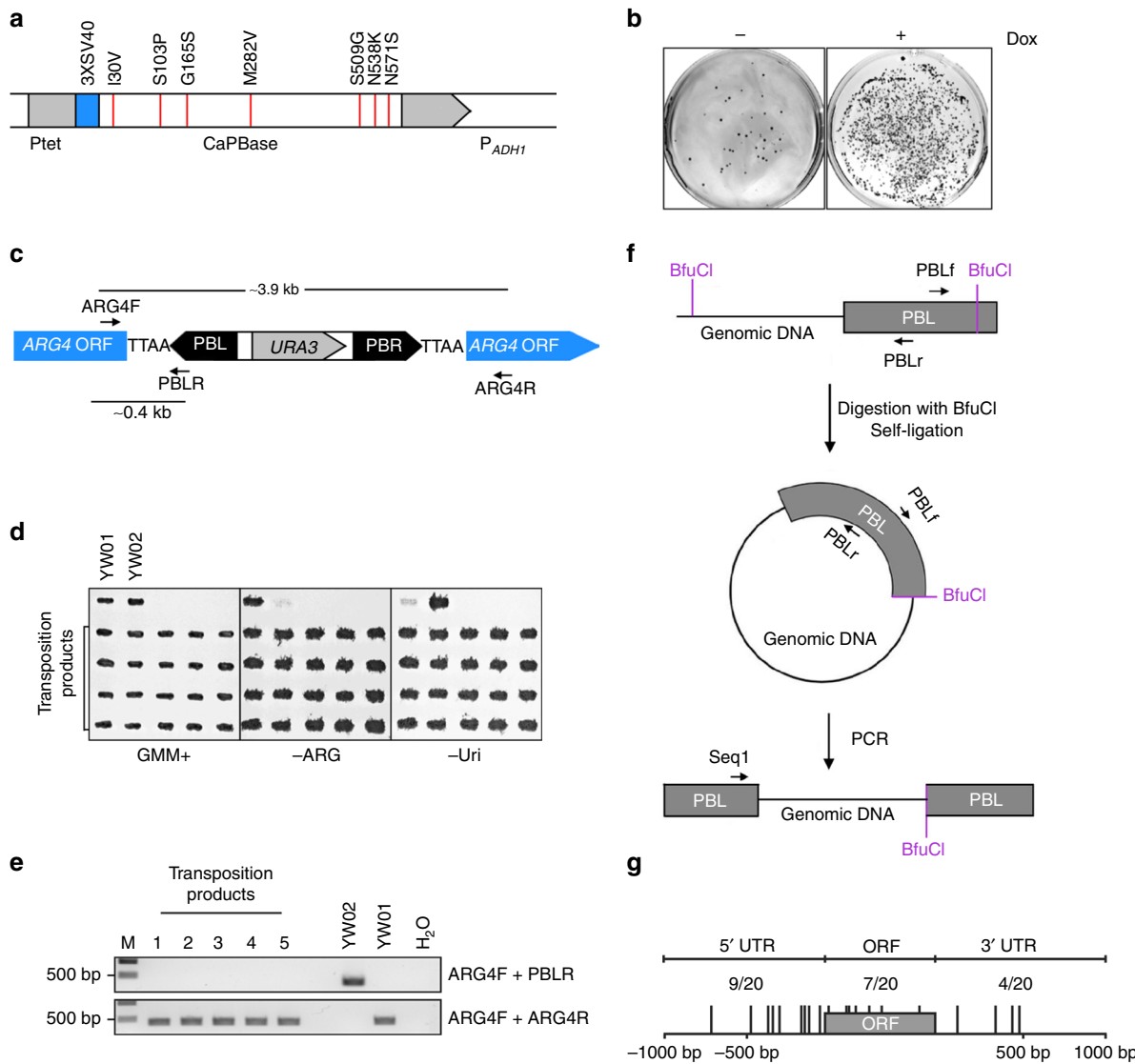

**Fig. 1** *PB* transposition in haploid *C. albicans*. **a** Schematic description of the CaPBase construct inserted at the *ADH1* locus. Ptet, Tet-On promoter; 3XSV40, SV40 nuclear localization signal; and P$_{ADH1}$, *ADH1* promoter. Hyperactive mutations are indicated along the top. **b** Selection of transformants with PB [URA3] integration on plates. YW01 cells were grown in YPD for 48 h in the presence (+) or absence (−) of Dox before transformation with pPB[URA3]. Transformation products were spread onto GMM plates (90 mm Petri dishes). **c** Schematic description of the PB[URA3] cassette integrated within *ARG4*. Arrows indicate the PCR primers used to detect *PB* excision. **d** Confirmation of the auxotrophic phenotypes of transposition products of YW02. Strains grown on a GMM+Uri+Arg+His (GMM+) plate were replica-transferred onto a GMM+Uri+His (-Arg) or GMM+Arg+His (-Uri) plate and then incubated at 30 °C overnight. YW01 and YW02 were included as controls. **e** Precise excision of *PB* from the *ARG4* locus. Representative gel electrophoresis results of PCR products amplified from YW02 transposition products using primers shown in **c**. **f** Schematic description of the identification of *PB* insertion sites by inverse PCR. Arrows indicate the PCR (PBLf and PBLr) and sequencing (Seq1) primers used. **g** Distribution of *PB* insertion sites in and around ORFs in YW02 transposition products. Positions in the ORF were determined by treating the entire length of an ORF as 100 and then determining the relative position of an insertion in the ORF. Positions in the 5′ or 3′ region were determined by whether the insertion was closest to the 5′ or 3′ end of an ORF

CaPBase. Tests in liquid medium produced similar results (Supplementary Figure 1a and 1b).

We next estimated *PB* transposition frequency. Aliquots of YW02 cells grown on solid medium containing Dox were harvested at intervals, and each was divided into two equal parts, and one was spread onto a YPD plate that allows all cells to grow and the other onto GMM+histidine (His) plates where only Arg +Ura+ transposants can grow. Transposition frequency was calculated by dividing the number of colonies on the GMM plate with that on the YPD plate. Figure 2c shows that *PB* transposition frequency increased with time during Dox induction, reaching

~5% at 72 h. We also achieved a similar frequency in liquid medium (Supplementary Figure 1c).

Multiple transposon insertions in a genome complicate subsequent identification of the mutation responsible for a phenotype. Although a single copy of *PB* transposon exists at the *ARG4* locus in YW02, the copy number may increase when transposition happens in S or G2 cells[23]. To test whether *PB* copy number changes during transposition, we analyzed the copy number of PB[URA3] in transposants. After 24 h of CaPBase induction, we spread the culture onto GMM+His plates to select transposants and randomly picked 10 colonies. We performed

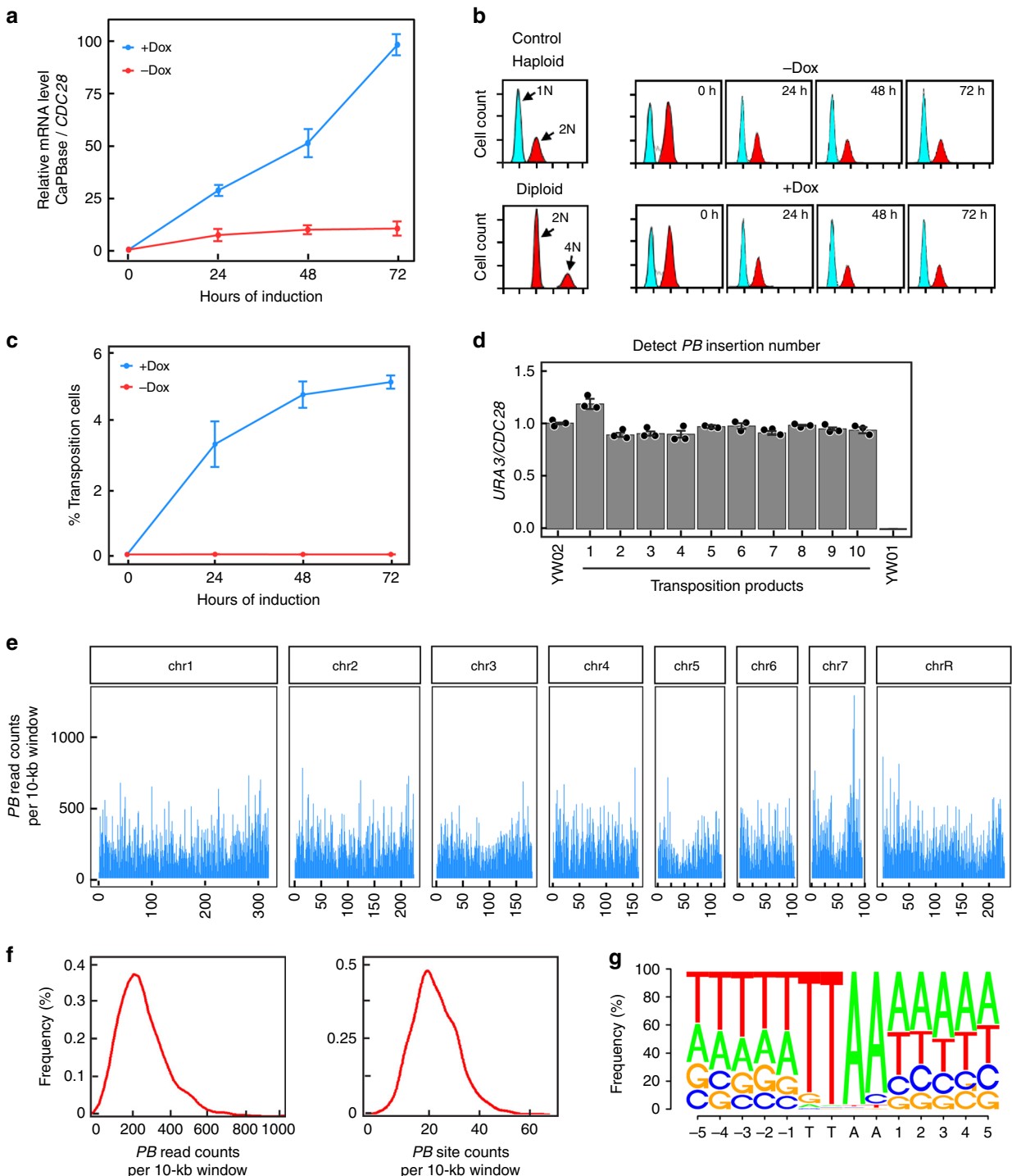

**Fig. 2** *PB* transposition and insert distribution in *C. albicans* genome. **a** qPCR analysis of CaPBase expression during transposition induction. *CDC28* mRNA level was used for normalization. Error bars represent standard deviation (s.d.) from the mean of triplicate samples. **b** Ploidy analysis during transposition induction by flow cytometry. The DNA content of induced cells at each time point was compared with that of haploid and diploid control strains. **c** Estimation of transposition efficiency in YW02. See the text for description. Error bars represent s.d. from the mean of three independent experiments. **d** Analysis of *PB* copy number in transposition products by comparing the levels of *URA3* and *CDC28* DNA using qPCR. YW01 and YW02 were included as controls. Error bars represent s.d. from the mean of triplicate samples. **e** Genome-wide NGS analysis of *PB* distribution. *PB*-specific reads were plotted in 10-kb sliding windows. **f** Histogram of *PB*-specific read counts and *PB*-specific site counts per 10-kb window. **g** Motif analysis of *PB*-insertion sites

qPCR to determine the level of *PB* DNA in comparison with a control gene *CDC28* in the genome. All clones were found to contain a single copy of *PB* (Fig. 2d and Supplementary Figure 1d), indicating that the vast majority of transposants harbored a single *PB* insertion in the genome.

**Genome-wide analysis of *PB* transposon distribution.** Next, we analyzed *PB* distribution in the genome using next-generation sequencing (NGS). Transposition was induced with Dox for 24 h, and then Arg+Ura+ transposants were selected on GMM+His plates. Genomic DNA was extracted from the transposant pool

and subjected to NGS. We acquired 11,595,368 overlapping read pairs, each containing one part matching the *PB* element and the other unambiguously matching a sequence of the *C. albicans* genome. The *PB* transposon distribution in 10-kb non-over-lapping windows is largely uniform throughout the genome (Fig. 2e and Supplementary Figure 1e). On average, there were 20 unique insertion sites in a 10-kb window (Fig. 2f), i.e., ~1 every 500 bp. We detected *PB* inserts in nearly 5000 genes out of a total of 6466 annotated and predicted genes in CGD. The failure to find *PB* insertion in the rest of the genes is likely due to their essentiality or the fitness cost caused by the *PB* insertion, consistent with the ~1100 essential genes in *S. cerevisiae*[24].

Nucleotide motif analysis of the insertion sites revealed 82.4% at TTAA sites, 4.5% at TTAG, 4.4% at CTAA, 3.6% at TTGT, and 5.1% at other non-specific sites (Fig. 2g). Also, we found significant enrichment of Ts and As in the 5 nucleotides immediately upstream and downstream of the *PB* inserts, respectively (Fig. 2g). *PB* insertion occurred more frequently in promoter regions than in ORFs. This could be due to the enrichment of TA dinucleotides in promoters, which facilitates the unwinding of double-stranded DNA during transcription initiation. Another reason could be the fitness cost caused *PB* insertion in some ORFs, which may reduce their presence in the transposant population.

**PB mutagenesis for forward genetic screens of 5-FOA resistant mutants**. To evaluate the effectiveness of the *PB* transposon system in genetic screens in *C. albicans*, we performed a proof-of-principle screen for mutants resistant to 5-fluoroorotic acid (5-FOA). As 5-FOA is converted to the toxic 5-fluorouracil in strains expressing a functional *URA3*[25], we wanted to determine how frequently we can identify *ura3* mutants in the transposon insertion library. We constructed a starting strain YW05 by using *URA3* to delete *HIS1* in GYZ803 and then inserting a PB[HIS1] cassette in *ARG4* ORF. YW05 also expresses CaPBase from the *ADH1* locus. Figure 3a describes the steps of the screen. YW05 cells were grown as patches on YPD plates containing Dox for 48 h to induce the transposition. Then, the cells were replica-transferred to GMM+Uri plates to enrich Arg+His+ transposants which were then transferred to GMM+Uri+5-FOA plates to select for 5-FOA-resistant mutants. Among 30 colonies randomly picked from the 5-FOA plates were 26 true 5-FOA-resistant mutants (Fig. 3b) and 4 clones (#3, #7, #18, and #21) were false and thus excluded from further characterization. Colony PCR using a pair of primers flanking *URA3* failed to detect the 1.4-kb amplicon expected from the WT strain in 22 of the 26 5-FOA-resistant mutants (Fig. 3c), suggesting PB[HIS1] insertion in *URA3*. *URA3* seemed intact in four clones, #4, #9, #15, and #26, suggesting *PB* disruption of other genes leading to 5-FOA resistance (Fig. 3c). Next, we performed PCR analysis of the 22 *ura3* mutants by pairing a *URA3*-flanking primer, URA3F or URA3R, with the *PB*-specific PBLR primer (Fig. 3d), followed by DNA sequencing analysis of the PCR products. We identified 4 different insertion sites in the *URA3* locus (Fig. 3e). One was located in the promoter region, and all the others were located in the ORF. Also, 19 *PB* insertions occurred at TTAA, one at TTAG, and two at CTAA motifs.

We also performed an NGS comparison of the mutant pools before and after 5-FOA treatment. We obtained 3,266,688 and 3,865,123 read pairs of *PB* insertion junctions from 5-FOA-treated and untreated samples respectively. Ranking the fold changes of genes after 5-FOA treatment, *URA5* and *URA3* occupied the first and second positions (Fig. 3f), exceeding the third by ~10 times. *URA5* and *URA3* also occupied the top eight places in the rank of fold changes of insertion sites after 5-FOA

treatment (Supplementary Figure 2a). Consistently, inverse PCR and sequencing analysis of 5-FOA resistant clones #4, #9, #15, and #26 confirmed PB[HIS1] insertion at the same position in *URA5* ORF (Fig. 3g). Deletion of *URA5* was found previously to cause partial resistance to 5-FOA in *S. cerevisiae*[25]. However, deleting *URA5* in both haploid and diploid *C. albicans* caused comparable 5-FOA resistance as deleting *URA3* (Supplementary Figure 2b). Ura3 and Ura5 catalyze two consecutive steps in the de novo biosynthesis of pyrimidines[26]. Taken together, this proof-of-principle screen demonstrated the superb power of this *PB*-mediated mutagenesis system in identifying genes required for a cellular process.

**Genome-wide genetic screening for fluconazole-resistant genes**. To identify new antifungal resistance mechanisms, we applied the *PB*-transposon mutagenesis system to perform genome-wide profiling of genes whose inactivation causes resistance to fluconazole. The rationale of the screen is: if *PB* insertion into a gene causes resistance to fluconazole, cells carrying this mutation will outcompete sensitive cells when grown in the presence of fluconazole and become overrepresented in the population. Thus, the subsequent NGS reads of *PB* insertions in this gene in the pooled genomic DNA will increase correspondingly. The *PB* insertion library was prepared by first growing YW02 cells in YPD+Dox medium and then dividing the culture into two equal parts for further cultivation in Dox-free medium with or without fluconazole. Cells were harvested after doubling 10 times and processed for NGS analysis.

We detected, in three independent experiments, an average of 5250 and 5122 genes from fluconazole-treated and untreated samples and ranked the genes according to the levels of enrichment after fluconazole treatment. Among the top 10 were three genes that either had been linked to azole susceptibility or are involved in ergosterol biosynthesis including *ERG3*, *ERG6*, and *ERG251* (Fig. 4a and Supplementary Data 2). We randomly selected 10 genes ranked higher than *PMC1*, which is the lowest ranking known resistant genes in our list, to construct haploid deletion mutants and found that they all showed increased resistance to fluconazole (Fig. 4b). Gene Ontology (GO) analyses of the genes that showed a significant increase in abundance in fluconazole-treated cells (log2 fold change > 1 and *p*-value < 0.05) revealed an overrepresentation of genes associated with ergosterol and sphingolipid biosynthesis (Fig. 4c). We also performed Gene Set Enrichment Analysis (GSEA) and found significant enrichment of genes associated with ergosterol and sphingolipid biosynthesis in fluconazole-treated cells in the leading edge subset that has the highest correlation with fluconazole resistance (Fig. 4d). The latter group included *FEN1*, *FEN12*, and *ARV1* that play important roles in sphingolipid biosynthesis and trafficking[27,28]. Like ergosterol, sphingolipids are the main components of the fungal plasma membrane which is both the target of azole and polyene classes of antifungal drugs and site of drug transporters. We reasoned that the discovery of these genes in our screen likely reflects some previously unknown mechanisms of drug resistance. Thus, we further characterized *FEN1* and *FEN12*, both encoding fatty-acid elongases that synthesize very-long-chain fatty acids as precursors of sphingolipid biosynthesis[28]. We verified that *fen1Δ/Δ* and *fen12Δ/Δ* mutants constructed in the diploid BWP17 background also exhibited significantly higher resistance to fluconazole than the WT strain (Fig. 4e).

**fenΔ/Δ mutants can tolerate higher levels of the toxic sterol**. Since blocking DMCDD synthesis by inactivating *ERG3* is a known resistance mechanism to fluconazole[29], we quantified

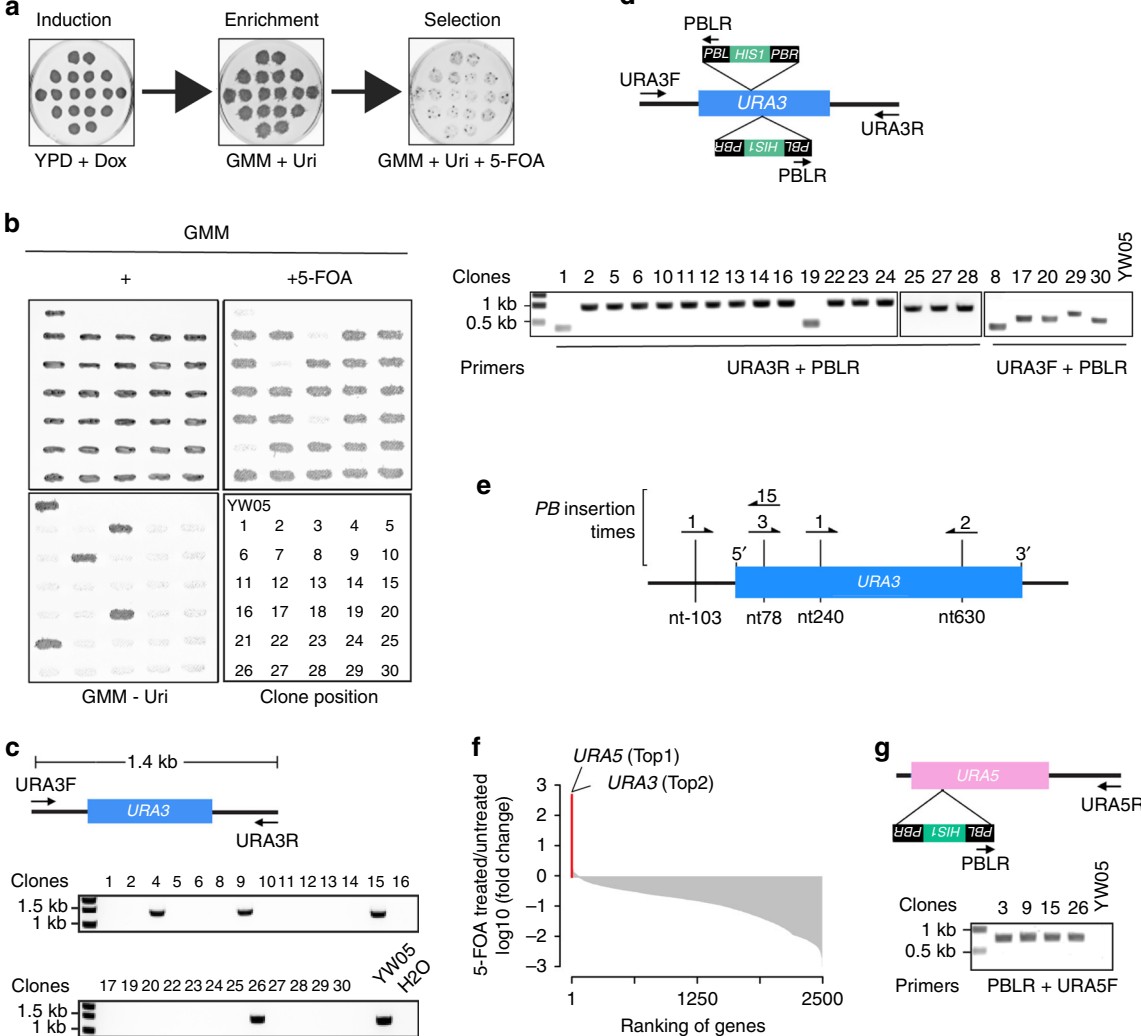

**Fig. 3** Genetic screen for 5-FOA-resistant mutants caused by *PB* transposition. **a** An illustration of the screening procedure. **b** Confirmation of the phenotype of 5-FOA-resistant mutants. Strains grown on a GMM+Uri+Arg+His (+) plate were replica-transferred onto a GMM+Uri+5-FOA (+5-FOA) or GMM+Arg+His (GMM−Uri) plate and then incubated at 30 °C overnight. YW05 was included as a control. The lower right panel shows the strain layout. **c** Characterization of 5-FOA-resistant mutants by colony PCR. Arrows indicate the primers used to detect *URA3*. **d** PCR identification of *PB* insertion sites at the *URA3* locus of the true 5-FOA-resistant clones from **b**. Arrows indicate the primers used. **e** *PB* insertion sites identified at the *URA3* locus. Sequence analysis identified four different sites indicated by vertical lines. Arrows denote *PB* insertions, and the direction of arrows indicate the direction of *HIS1* transcription in *PB*. The numbers indicate the number of mutants associated with each insertion site. **f** Fold changes of genes with *PB* inserts after growing the mutant library in media containing 5-FOA in comparison with the same library grown in normal media. The top two genes are highlighted in red. **g** PCR identification of *PB* insertion sites at the *URA5* locus. Arrows denote the primers used

ergosterol and DMCDD in whole cell lysates and the plasma membrane using LC-MS. Due to the lack of standard DMCDD, we included an *erg3Δ* strain to ensure correct identification of this molecule by combining the detected molecular mass and the absence of the compound in *erg3Δ* cells (Fig. 5a, top and Supplementary Figure 3). LC-MS detected a ~90% decrease of ergosterol in the whole cell lysates of both *fen1Δ/Δ* and *fen12Δ/Δ* mutants compared to WT controls grown in drug-free media (Fig. 5a, middle), indicating that lacking either *FEN1* or *FEN12* impairs ergosterol biosynthesis severely. As expected, fluconazole treatment caused a dramatic drop of ergosterol levels in all strains. Unexpectedly, we detected a ~270% increase in the amount of DMCDD in both *fen1Δ/Δ* and *fen12Δ/Δ* cell lysates compared with WT cells upon fluconazole treatment (Fig. 5a, bottom). However, the DMCDD level in the plasma membrane fraction of *fen1Δ/Δ* and *fen12Δ/Δ* cells was found to be 35 and 29% of that in WT cells (Fig. 5a, bottom). The results suggest that,

unlike *erg3Δ/Δ*, the higher fluconazole resistance of *fenΔ/Δ* mutants is not due to decreased production of the toxic sterol.

**Significant changes in sphingolipid composition in *fenΔ/Δ* mutants.** As ergosterol and DMCDD insert themselves into the cell membrane mainly through interaction with sphingolipids[30], we speculated that changes of the sphingolipid profile in *fen1Δ/Δ* and *fen12Δ/Δ* might be responsible for the tolerance of DMCDD. To test this hypothesis, we performed lipidomics profiling by LC-MS. We detected rather different fatty-acid profiles between *fen1Δ/Δ* and *fen12Δ/Δ* mutants (Fig. 5b and Supplementary Figure 4). The levels of fatty acids with 24 and 26 carbons decreased by 74–85% in both mutants grown in drug-free media (Fig. 5b), consistent with the loss of the very-long-chain fatty acid elongase. Varying degrees of increase or decrease were observed in the levels of fatty acids with 14–22 carbons; for instance, *fen1Δ/Δ* cells grown in drug-free media showed 10–60% decrease in

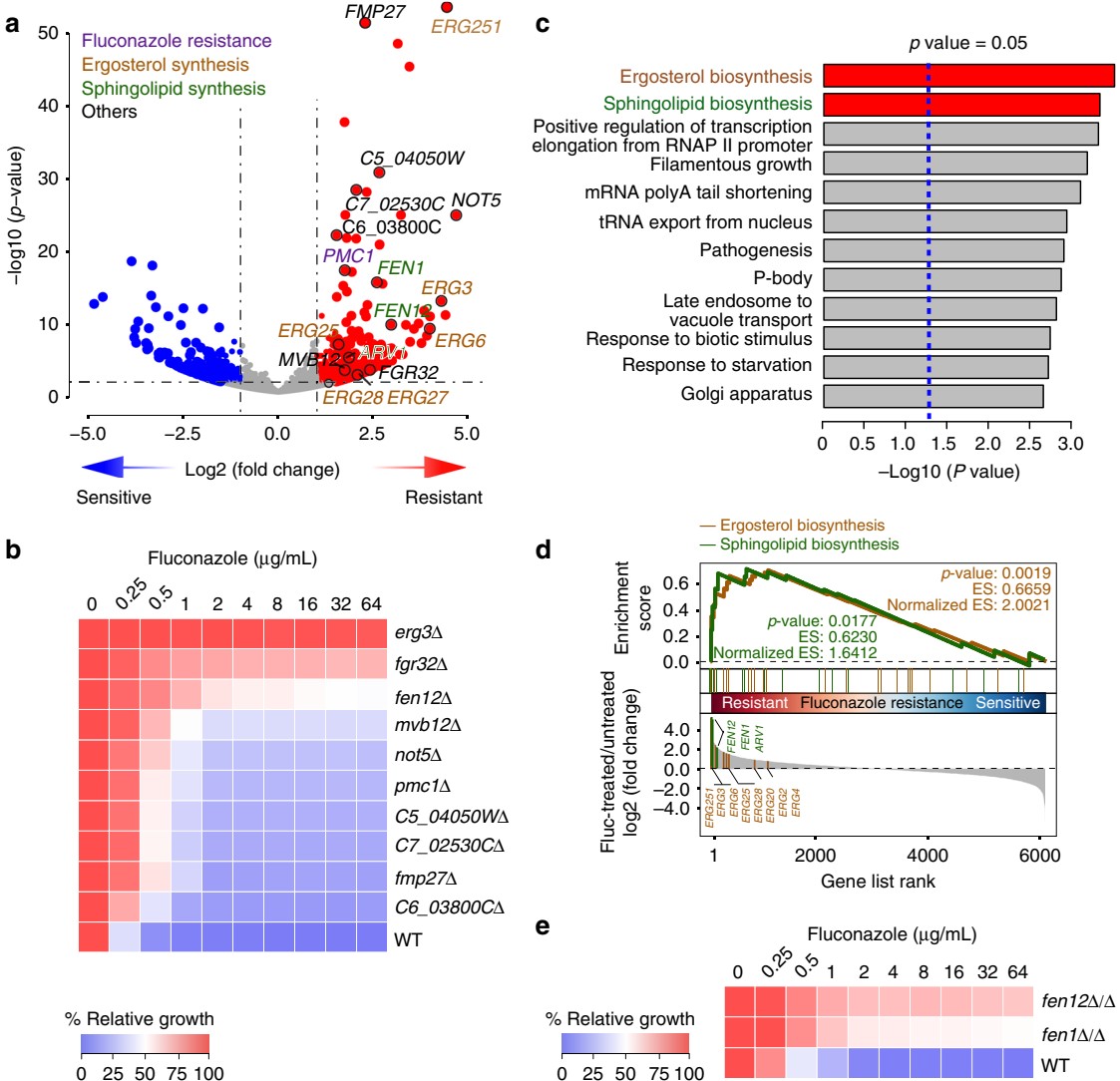

**Fig. 4** Genetic screens for fluconazole-resistant mutants. **a** Volcano plot of fold changes of genes with *PB* inserts in *PB* transposition mutants grown in media containing 100 μg/mL fluconazole in comparison with the same mutants grown in normal media. *p*-Values were calculated using the Wald test. Ten candidates selected for phenotype confirmation are indicated. Genes known to be associated with fluconazole resistance and ergosterol and sphingolipid biosynthesis are shown with different colors. **b** Gene deletion of 10 candidates randomly selected from the top-ranked genes in **a** confirms increased resistance to fluconazole. Fluconazole susceptibility was evaluated using the standard CLSI broth microdilution protocol[67]. Cultures were incubated for 24 h, and growth was measured as OD$_{600}$ and expressed as a percentage of the growth in the control wells with no drug (DMSO alone) for each strain. **c** Gene Ontology (GO) analysis of the *PB*-mutated genes that led to statistically significant fold increases (*p* < 0.05) after fluconazole treatment. Red bars highlight the ergosterol and sphingolipid biosynthesis pathway. *p*-Values were from Fisher's exact test. **d** Gene Set Enrichment Analysis (GSEA) of the NGS data from **a**. *p*-Values were determined using the Kolmogorov–Smirnov test. **e** Homozygous deletion of *FEN1* or *FEN12* in BWP17 also results in increased fluconazole resistance. The assay was done as described in **b**

these fatty acids while *fen12Δ/Δ* cells showed 164% and 419% increase in FA(20:0) and FA(22:0), respectively (Fig. 5b). Similar fatty-acid profiles were detected in fluconazole-treated cells. Consistently, we detected a marked decrease in the levels of α-hydroxyphytoceramide (αHPC), inositolphosphoceramide (IPC), and mannosylinositolphosphoceramide (MIPC) species with a 26-carbon fatty acid chain in both *fen1Δ/Δ* and *fen12Δ/Δ* cells grown in drug-free media (Fig. 5b and Supplementary Figures 5, 6 and 7); in particular, there was a 20-fold decrease of d18:0/26:0 αHPC and ~50-fold decrease of both d18:0/26:0 IPC and d18:0/26:0 MIPC in *fen12Δ/Δ* cells. In contrast, several αHPC, IPC, and MIPC molecules with a 20–24-carbon fatty acid chain showed an increase over a broad range; for example, d16:0/24:0 αHPC increased by 37.4 fold, 18:0/22:0 or d16:0/24:0 (the two molecules

have identical mass) IPC by 73.3 fold, and d18:0/22:0 or d16:0/24:0 MIPC by almost 300 fold in *fen12Δ/Δ* cells (Fig. 5b). Strikingly, in response to fluconazole treatment, all species of αHPC and MIPC increased significantly in WT cells from 5.48 to 99.3 fold. This fluconazole-induced increase was more dramatic in *fenΔ/Δ* cells; for example, d16:0/20:0 αHPC increased by 50.5 fold in *fen1Δ/Δ* and d16:0/24:0 αHPC increased by 361 fold in *fen12Δ/ Δ* cells. The most significant changes occurred in d18:0/22:0 or d16:0/24:0 and d18:0/20:0 or d16:0/22:0 MIPC which increased by >3 orders of magnitude (Fig. 5b). The results suggest that increasing the cellular levels of some species of sphingolipids might be a natural stress response to fluconazole. We speculate that high levels of sphingolipids could improve cell membrane strength leading to higher tolerance of the toxic sterol.

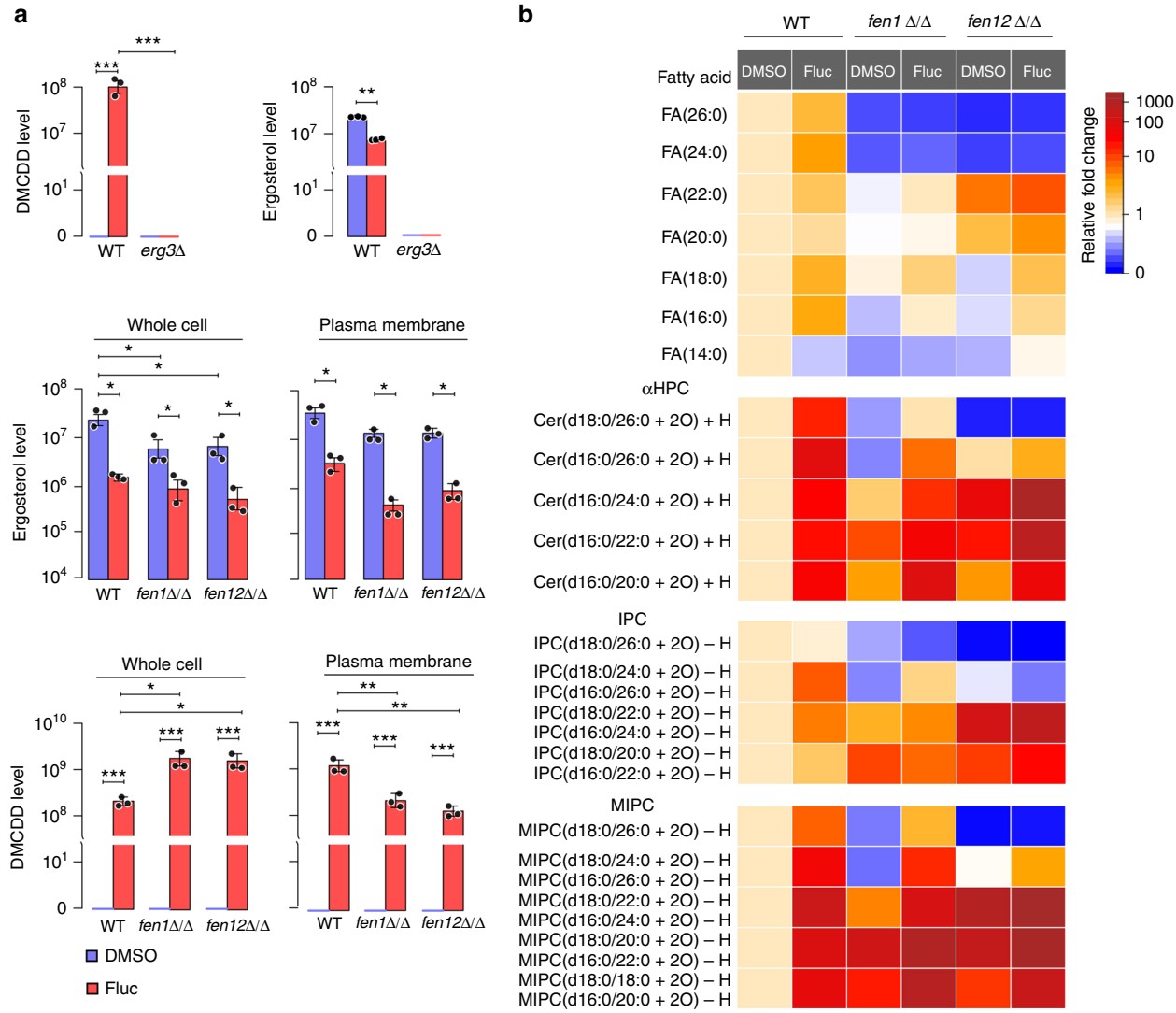

**Fig. 5** LC-MS analysis of sterols and lipids in WT and *fenΔ/Δ* cells. **a** LC-MS comparison of ergosterol and DMCDD levels in whole cell lysates and the plasma membrane fraction of WT (GZY803 and BWP17), *erg3Δ*, *fen1Δ/Δ*, and *fen12Δ/Δ* cells with or without fluconazole treatment. Error bars represent s.d. from the mean of three independent experiments. Significance was measured using unpaired *t*-test (*$p < 0.05$, **$p < 0.01$, ***$p < 0.001$). **b** LC-MS analysis of free fatty acids, αHPC, IPC, and MIPC sphingolipids in WT, *fen1Δ/Δ*, and *fen12Δ/Δ* cells with or without fluconazole treatment

**Mechanisms of sphingolipid overproduction in *fenΔ/Δ* mutants.** We further investigated the mechanisms behind the fluconazole-induced sphingolipid overproduction. Fungal cells possess a remarkable capacity to counterbalance a disturbance to maintain cellular homeostasis. In response to fluconazole treatment, *C. albicans* reacts by overexpressing *UPC2* which encodes a transcription factor that regulates ergosterol biosynthesis (Fig. 6a summarizes the biosynthetic pathways of ergosterol and sphingolipids) and sterol uptake[31]. Indeed, we detected the overexpression of *UPC2*, *ERG3*, and *ERG11* in both WT and *fenΔ/Δ* strains after fluconazole treatment (Fig. 6b). *UPC2* also regulates genes involved in sphingolipid biosynthesis[32–35]. Consistently, we detected overexpression of the ceramide synthase gene *LAG1*, the IPC synthase gene *AUR1*, the MIPC synthase gene *MIT1*, and the M(IP)₂C synthase gene *IPT1* upon fluconazole treatment (Fig. 6b). Importantly, the fluconazole-induced *UPC2* overexpression is much stronger in *fenΔ/Δ* cells than in WT cells. The results are consistent with the higher amounts of sphingolipids found in the mutants grown in the presence of fluconazole.

To determine whether the upregulation of sphingolipid biosynthesis is responsible for the increased resistance to fluconazole in *fen1Δ/Δ* and *fen12Δ/Δ* mutants, we attempted to delete *UPC2* and *LAG1* in *fen1Δ/Δ* and *fen12Δ/Δ* strains. Despite repeated efforts, we could not delete both copies of *LAG1* in either mutant and those of *UPC2* in *fen1Δ/Δ*, suggesting synthetic lethality. However, we found that deleting one copy of *LAG1* or one copy of *UPC2* in *fen1Δ/Δ* and *fen12Δ/Δ* appeared to increase their sensitivity to fluconazole (Fig. 6c). Also, deleting both copies of *UPC2* in *fen12Δ/Δ* abolished its resistance to fluconazole (Fig. 6c). The results suggest that Upc2-induced upregulation of sphingolipid biosynthesis may underlie the increased fluconazole resistance of *fen1Δ/Δ* and *fen12Δ/Δ* mutants. Consistently, the fluconazole-induced upregulation of *ERG3*, *ERG11*, *LAG1*, *AUR1*, *MIT1*, and *IPT1* was significantly reduced or abolished when one or both copies of *UPC2* was deleted (Fig. 6b). Together, the results corroborate the hypothesis that *fen1Δ/Δ* and *fen12Δ/Δ* mutants increase the cellular levels of sphingolipids in response to fluconazole by strongly upregulating the expression of genes

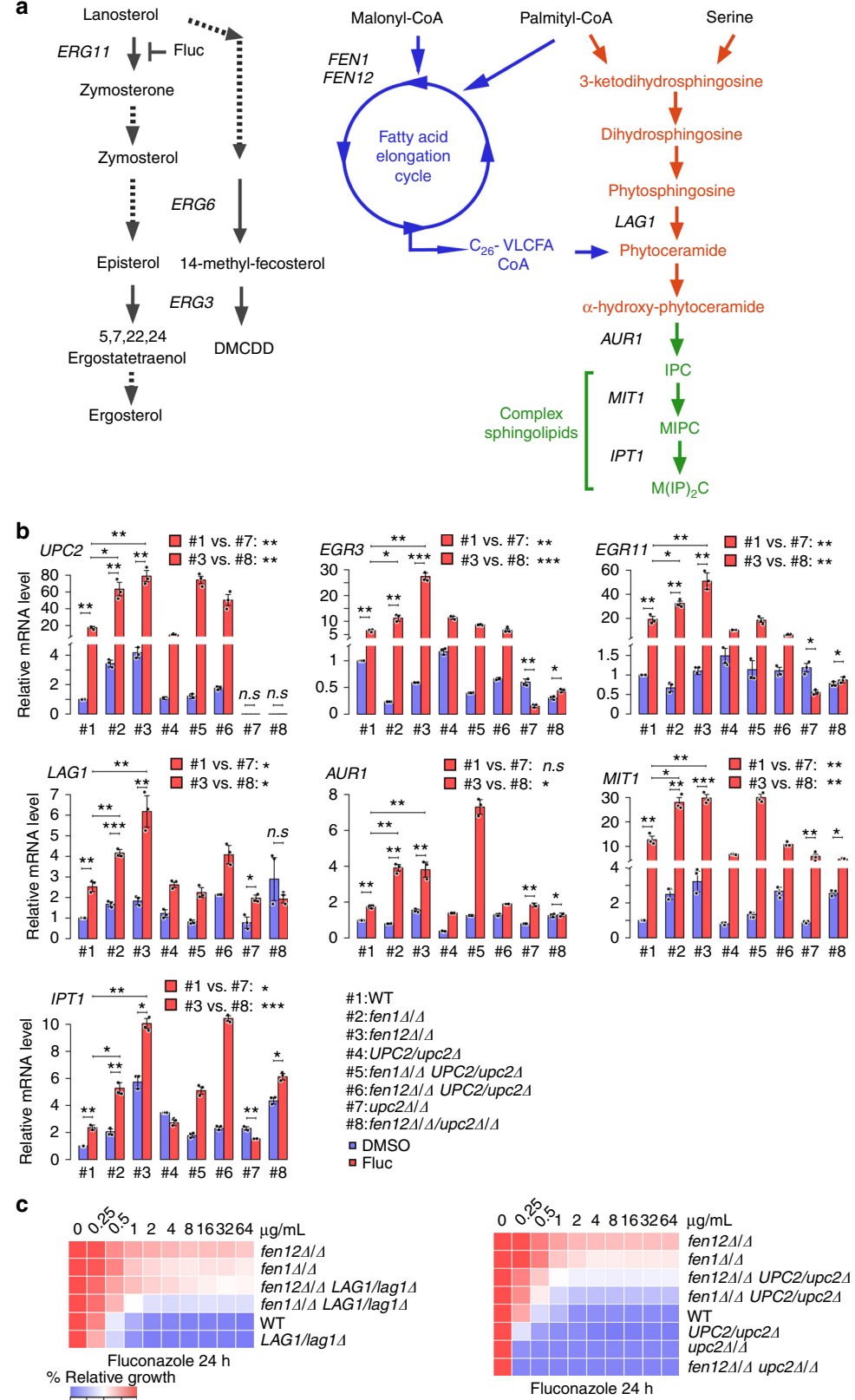

**Fig. 6** Sphingolipid overproduction underlies fluconazole resistance of *fenΔ/Δ* cells. **a** Illustration of ergosterol and sphingolipid biosynthesis pathways. **b** qPCR analysis of *UPC2*, *ERG3*, *ERG11*, *LAG1*, *AUR1*, *MIT1*, and *IPT1* expression in WT, *fen1Δ/Δ*, and *fen12Δ/Δ* mutants with *UPC2* single or double knockout in the presence or absence of fluconazole. Error bars represent s.d. from the mean of three independent experiments. Significance was measured using unpaired *t*-test (*$p < 0.05$, **$p < 0.01$, ***$p < 0.001$, n.s., not significant). **c** Fluconazole susceptibility of WT and the indicated mutants was evaluated using the standard CLSI broth microdilution protocol. Following 24 h of incubation, growth was measured as $OD_{600}$ and expressed as a percentage of the growth in the control wells with no drug (DMSO alone) for each strain

involved in sphingolipid biosynthesis, leading to tolerance of the toxic sterol.

## Discussion

In recent years, considerable efforts have been made to introduce transposon mutagenesis strategies to create libraries of heterozygous[36,37] or homozygous[38,39] mutants in *C. albicans*. In general, these methods start with in vitro transposition of a transposon-containing cassette, Tn5-UAU1 or Tn7-UAU1, into a *C. albicans* genomic DNA library[36,37]. Then, the mutagenized library is transformed into *Escherichia coli* for amplification before transformation into *C. albicans*. These methods are tedious, labor-intensive, and unsuitable for high-throughput genome-wide genetic studies. Also, their use in diploid *C. albicans* creates heterozygous mutations that do not result in readily detectable phenotypes except for a small number of haploinsufficient genes[40,41]. Mielich et al. recently introduced the *Activator/Dissociation* transposon of maize into haploid *C. albicans*[42], but the system is not very useful at its current state because of frequent autodiploidization.

Here, we introduce a simple and efficient *PB* transposon-based mutagenesis system in haploid *C. albicans*, which has the following advantageous features. First, a single transposon insertion in the promoter or ORF of a gene is, in most cases, sufficient to inactivate a gene and produce a phenotype. Second, the transposon donor is inserted into a selectable marker gene, i.e., *ARG4*, and the transposase is expressed from a different locus under the control of the Tet-On promoter. Thus, a mutant library can be generated merely by growing the cells in a medium containing Dox to express the transposase and then selecting Arg+ transposants in -Arg media. Third, transposition occurs at high frequencies and in a random manner. Our NGS analysis showed an even distribution of *PB* insertions on all chromosomes and identified *PB* inserts in ~5000 genes (Fig. 2e). Fourth, each mutant cell contains a single *PB* insertion, which allows straightforward mapping of the affected gene by inverse PCR and DNA sequencing. Fifth, the precise excision of *PB* from the insertion site restores the coding sequence allowing easy confirmation of the phenotype–genotype relationship. Our successful identification of *URA3* and *URA5* in 26 out of 30 random 5-FOA-resistant mutants demonstrates the superb power of the *PB* transposon mutagenesis system (Fig. 3). Furthermore, the *PB*-mediated mutagenesis system is readily applicable to whole-genome epistasis analysis of gene networks by first deleting a gene of interest before activating the transposition and screening for synthetic phenotypes. We anticipate that this *PB* transposon-based mutagenesis platform will significantly accelerate discoveries to advance our understanding of the biology and pathogenesis of this pathogen.

Frequent isolation of multidrug-resistant *Candida* worldwide is a cause for concern because currently only three classes of drugs are available approved for the treatment of systemic fungal infections[43,44]. Hard lessons learned since the discovery of penicillin have taught us that microbial pathogens are extremely innovative in quickly finding tricks to defeat every one of our therapies. To win the war against infectious disease, it is imperative to elucidate how a pathogen has acquired resistance and even to predict how it can do it. The development of the *PB*-transposon mutagenesis system has allowed us to conduct genome-wide screening in *C. albicans* to identify genes underlying antifungal susceptibility. Our NGS analyses of *PB* transposon insertion sites in mutants selected by growth in the presence of fluconazole in comparison with mutants grown in drug-free media revealed significant enrichment of genes involved in ergosterol biosynthesis, in particular, *ERG3* and *ERG6*

whose mutation is well known to increase fungal resistance to azoles[45–48]. Deleting any one of 10 genes randomly picked from our resistant mutant list all led to increased fluconazole resistance (Fig. 4b), indicating that the screen was successful. Among the highest ranking genes were *FEN1*, *FEN12*, and *ARV1* which play important roles in sphingolipid biosynthesis[49,50]. The understanding that sphingolipids are major components of fungal cell membrane and interact both physically and functionally with ergosterol[30,51,52], the target of azole and polyene antifungal drugs, prompted us to conduct a comprehensive analysis of *FEN1* and *FEN12*. Our first informative finding was a severalfold increase in the cellular levels of the toxic sterol in both *fen*Δ/Δ mutants compared to WT cells (Fig. 5a), which indicated a resistance mechanism distinct from that of the *erg3* mutant[2,46]. Interestingly, the level of the toxic sterol in the plasma membrane is significantly lower in the mutants than in the wild type (Fig. 5a), suggesting that the mutants pack fewer toxic sterol molecules into the membrane. Furthermore, we detected dramatic changes in the sphingolipid composition in whole cell extracts (Fig. 5b). The most striking change is the increase of more than 3 orders of magnitude of some MIPC species in *fen12*Δ/Δ cells upon fluconazole treatment, which also increased the total cellular level of sphingolipids. An illuminating observation was the sharp increase of the same MIPC species in WT cells after fluconazole treatment (Fig. 5b), indicating that *C. albicans* alters its sphingolipid composition as a natural stress response mechanism to the disruption of ergosterol biosynthesis. This could be a conserved survival strategy because eukaryotic cells are known to have a huge capacity to preferentially modify the cellular levels and make-up of sphingolipids in response to changes in sterol composition in their membrane as an adaptation mechanism to maintain membrane structure and function[52]. In *S. cerevisiae*, mutation of *ELO3*, a homolog of *FEN12*, also affected sphingolipid composition significantly[53]. Many previous studies have revealed intricate relationships between ergosterol and sphingolipid biosynthesis and resistance to azoles. Particularly consistent with our findings, Prasad et al. found that blocking M(IP)$_2$C synthesis by deleting the *IPT1* gene significantly increased *C. albicans* susceptibility to azoles[54]; furthermore, the level of the drug efflux pump Cdr1 in the plasma membrane was reduced in *ipt1*Δ/Δ cells as well as in cells treated with fumonisin B1, an inhibitor which blocks the synthesis of phytoceramide[55]. Consistently, we demonstrated that blocking sphingolipid biosynthesis reduced the fluconazole resistance in *fen1*Δ/Δ and *fen12*Δ/Δ cells. Sphingolipids are abundant components of the fungal cell membrane and play both structural and signaling roles[56]. Numerous experiments have shown that changes of sphingolipid composition can dramatically alter the properties of cell membranes, such as rigidity, fluidity, and domain organization which further influences cellular functions[57]. Thus, the marked changes of the cellular sphingolipid composition in *fen*Δ/Δ mutants might lead to fluconazole resistance by reducing the incorporation of the toxic sterol into the cell membrane and limiting subsequent structural and functional damage of the cell membrane. Although we do not have direct evidence confirming repellence of the toxic sterol from the cell membrane by high levels of sphingolipids, this can be expected based on the chemical properties of sterols and sphingolipids and is indirectly supported by various previous studies. Cell membranes have a complex organization featuring domains with distinct lipid compositions known as rafts whose formation is determined by the hydrophobic interaction between sterols and sphingolipids[58,59]. However, the toxic sterol is structurally different from ergosterol in terms of the absence of B-ring unsaturation, the presence of the 6-OH group and the additional C-14 methyl group, and side-chain modification. In particular, the polar 6-OH group is thought to interfere with sterol–sphingolipid

packing in the plasma membrane, causing membrane damage[60,61]. The high level of sphingolipids in $fen\Delta/\Delta$ mutants could strengthen the cell membrane by restraining membrane deformation. When the additional free energy required for incorporating polar groups of the toxic sterol cannot be compensated by membrane deformation, they must be pushed out to reach a lower energy status. Thus, more toxic sterol molecules with a polar side chain would be repelled from the membrane by the nonpolar chain of sphingolipids. Taken together, it is reasonable to believe that high levels of sphingolipids can repel toxic sterol molecules from the cell membrane. It was reported that $fen1\Delta/\Delta$ and $fen12\Delta/\Delta$ mutants exhibited increased sensitivity to amphotericin B[62]. It is widely accepted that amphotericin B binds to ergosterol in a parallel manner to form barrel-stave type pores and penetrate cell membranes, in which their hydrophilic polyhydroxy side is pointing inward to constitute the pore lining and their hydrophobic lipophilic heptaene part is directing outward to interact with the membrane interior[63,64]. While the high levels of sphingolipids in the membrane hinder the insertion of the hydrophilic toxic sterol, the binding of amphotericin B to ergosterol is not affected; and on the contrary, the shorter sphingolipids in $fen\Delta/\Delta$ mutants could expose ergosterol molecules and thus facilitate the binding to amphotericin B, making mutant cells more sensitive to this drug. In future studies, it would be interesting to know whether high cellular levels of sphingolipids contribute to azole resistance in clinical isolates of *C. albicans* and other fungal pathogens.

## Methods

**Strains, media, and growth conditions**. All haploid *C. albicans* strains were derived from GZY803 (*MTLα his4 ura3Δ::HIS4*)[14]. YW02 (*MTLα his4 ura3Δ::HIS4* PTet-On-CaPBase::*SAT1 arg4::PB[URA3]*), and YW05 (*MTLα his4 ura3Δ::HIS4 his1Δ::URA3* PTet-On-CaPBase::*SAT1 arg4::PB[HIS1]*) were used as the parental strain for transposon mutagenesis. A complete list of strains and their genotypes are provided in Supplementary Table 2. *C. albicans* was routinely grown at 30 °C in YPD medium (2% yeast extract, 1% bactopeptone, and 2% glucose) or in GMM (2% glucose and 6.79 g/L yeast nitrogen base without amino acids). 5-FOA (Sigma, F5013-250MG) was used at a concentration of 1 g/L in GMM plates supplemented with uridine. Dox (Sigma, D9891-1G) was used in YPD medium at a final concentration of 50 µg/mL to induce the expression of CaPBase. Stock solutions of fluconazole (Sigma, F8929-100MG) was prepared in dimethyl sulfoxide (Sigma, V900090-500ML) and stored at −20 °C until use.

**Plasmid construction**. pPB[URA3] was constructed as follows: the *URA3* marker was PCR-amplified from a pBluescript-based plasmid with primers URA3ApaI and URA3SacII (all primers used in this study are shown in Supplementary Data 1), and cloned into the *ApaI* and *SacII* sites of pPB[ura4] to generate pPB[URA3].

Construction of pARG4-PB[URA3]: the *EcoRI-PstI* fragment including the intact PB[URA3] transposon cassette was PCR-amplified from pPB[URA3] with primers PB[URA3]EcoRI and PB[URA3]PstI, and inserted into the *EcoRI* and *PstI* sites of a pBluescript-based plasmid containing *ARG4* selectable marker.

Construction of pARG4-PB[HIS1]: the *HIS1* marker was PCR-amplified from a pBluescript-based plasmid with primers HIS1ApaI and HIS1SalI, and cloned into the ApaI and SalI sites of pARG4-PB[URA3] to generate pARG4-PB[HIS1].

Construction of pNIM1-CaPBase: site-directed mutagenesis of the PBase coding sequence in plasmid pDUAL-PBase was performed using Quick-Change Site-Directed Mutagenesis Kit (Stratagene, 200514) to generate pDUAL-CaPBase which contains a codon-optimized PBase with hyperactivity. Then a PCR fragment containing the CaPBase was amplified from pDUAL-CaPBase with primers PBaseXhoI and PBaseBamHI, and cloned into the *SalI* and *BglII* sites of pNIM1 to generate pNIM1-CaPBase. pPB[ura4], pDUAL-PBase, and pNIM1 were described previously[20,65].

**Construction of strains YW02 and YW05**. YW02 was constructed from GZY803 by introducing the linear DNA fragment from pNIM1-CaPBase containing the PTet-On-CaPBase fusion and the XhoI-NotI fragment from pARG4-PB[URA3] containing the ARG4-PB[URA3] fusion to transform GZY803 to become nourseothricin resistance and uridine prototrophic. Correct integration was confirmed by PCR.

YW05 was constructed from YW11 by introducing the linear DNA fragment from pNIM1-CaPBase containing the PTet-On-CaPBase fusion and the XhoI-NotI fragment from pARG4-PB[HIS1] containing the ARG4-PB[HIS1] fusion to transform YW11 to become nourseothricin resistance and histidine prototrophic. Correct integration was confirmed by PCR.

**Inverse PCR for mapping *PB* insertion sites**. Genomic DNA was prepared using MasterPure Yeast DNA Purification Kit (Epicentre Technologies, MPY80200) and digested with BfuCI overnight. After heat inactivation of BfuCI, the digested genomic DNA was self-ligated with T4 DNA ligase at 16 °C overnight. Primers used to recover the flanking sequence of the left side of the *PB* transposon were PBLf and PBLr. PCR products were purified from agarose gel and sequenced using Seq1 as the sequencing primer. The sequencing results were used to BLAST-search *Candida* Genome Database ([www.candidagenome.org](http://www.candidagenome.org)) to determine the *PB* insertion sites.

**Quantitative PCR**. To determine the copy number of *PB[URA3]*, genomic DNA was purified using MasterPure yeast DNA purification kit and analyzed on STRATAGENE Mx3000p using Maxima SYBR Green qPCR Master Mix (2X) with separate ROX vial (Thermo Fisher Scientific, K0253) according to the manufacturer's instructions. PCR was carried out with primers CDC28 sense and CDC28 antisense to amplify *CDC28*, URA3 sense and URA3 antisense to amplify *PB [URA3]*. *PB* copy number was determined by ΔΔCT method using the genomic DNA of strain YW02 as a control, which has a single copy of *PB*[66]. Since the ratios of *PB* versus *CDC28* were usually not integral numbers, we estimated the copy number to be the integral closest to the ratios.

To detect the expression of CaPBase, *ERG11*, *ERG3*, *UPC2*, *LAG1*, *AUR1*, *MIT1*, and *IPT1*, total RNA was isolated using the RNeasy Mini Kit (QIAGEN, 74106) and RNase-Free DNase Set (QIAGEN, 79254) and then reverse-transcribed using the SuperScript™ II Reverse Transcriptase (Invitrogen, 18064014). Changes in transcript levels of target genes were analyzed using the Maxima SYBR Green qPCR Master Mix (2X) with separate ROX vial with primers PBase sense and PBase antisense for CaPBase, ERG11 sense and ERG11 antisense for *ERG11*, ERG3 sense and ERG3 antisense for *ERG3*, UPC2 sense and UPC2 antisense for *UPC2*, LAG1 sense and LAG1 antisense for *LAG1*, AUR1 sense and AUR1 antisense for *AUR1*, MIT1 sense and MIT1 antisense for *MIT1*, IPT1 sense and IPT1 antisense for *IPT1*, and normalized to *CDC28* transcript levels using the ΔΔCT method.

**Transposition induction**. In the experiments using integrated PB[URA3] or PB [HIS1] as the transposon donor, a single colony of YW02 or YW05 was first cultured in liquid YPD medium at 30 °C overnight. Then, the cells were transferred to liquid or solid YPD medium supplemented with 50 µg/mL Dox to induce the expression of CaPBase and undergo transposition. To calculate transposition efficiency, aliquots of cells were harvested at timed intervals and spread on GMM plates to count the number of cells that had undergone transposition and on YPD plates to obtain the total number of cells. Transposition efficiency was calculated as the number of colonies on GMM plates divided by the number of colonies on YPD plates.

**Flow cytometry**. Flow cytometry analysis was performed using a BD FACSCalibur as described previously[14,22]. Mid-log phase cells were collected, washed, and fixed with 70% (vol/vol) ethanol (Sigma, E7023-500ML). Cells were washed with solution I (200 mM Tris–HCl, pH 7.5 and 20 mM EDTA) once and treated with 10 mg/mL RNase A (Sigma, R4875-100MG) (1:100 dilution in solution I). Cells were washed with PBS and resuspended in 5 mg/mL propidium iodide (Sigma, 81845-25MG) (1:50 dilution in PBS) incubated in the dark at 4 °C overnight. Stained cells were collected and resuspended in PBS and sonicated before loading.

**Mapping *PB* insertion sites by next-generation sequencing**. A single colony of YW02 or YW05 cells were inoculated into 5 mL of YPD and grown at 30 °C to the exponential phase. Eighteen equal aliquots of the culture, each containing about 10,000 cells, were spotted on a YPD plate containing 50 µg/mL Dox to induce CaPBase expression. After 24 h of incubation at 30 °C, cells on the induction plates were replica-transferred to a GMM plate without Dox to enrich the cells that had undergone transposition. The GMM plate was incubated at 30 °C for 2 days before collecting the cells for the extraction of genomic DNA using the MasterPure Yeast DNA Purification Kit. Tagmentation was carried out by using the TruePrep DNA Library Prep Kit V2 for Illumina (Vazyme Biotech, TD501) according to the manufacturer's instructions. The first round of PCR was carried out to amplify *PB* together with its flanking sequences using a *PB*-specific primer close to the insertion site, PBseq and a primer annealing to the Tn5 adaptor, N7XX. The cycling parameters were: 3 min at 72 °C; 30 s at 98 °C; 12 cycles of 15 s at 98 °C, 30 s at 60 °C, and 3 min at 72 °C; 10 min at 72 °C; 4 °C forever. The second round of PCR was performed using NEBNext® Ultra™ II Q5® Master Mix (NEB, M0544L) to add sequences required for Illumina sequencing with an Illumina flowcell attachment primer, TruPBseq and partial Tn5 adaptor primer, Illumina-R. Cycling conditions are: 1 min at 98 °C; 7 cycles of 10 s at 98 °C, 75 s at 67 °C; 5 min at 67 °C; 4 °C forever. The size selection and purification of the PCR products were carried out with AMPure XP beads (Beckman Coulter, A63882).

**Genetic screens using *PB*-mediated mutagenesis system**. In the screen for 5-FOA-resistant mutants, insertional mutagenesis in YW05 cells was induced by culturing on YPD plate containing Dox for 24 h and then replica-transferred to GMM plates containing uridine to select for Arg+His+ cells. After incubation at 30 °C for 24 h, cells on the above GMM plates were replica-transferred to GMM plates supplemented with uridine and 5-FOA. The 5-FOA plates were incubated at 30 °C until resistant colonies had formed. For screening in the liquid medium, Arg+His+ mutant cells were collected and washed three times with PBS. Two equal aliquots of cells were cultivated further in 20 mL GMM medium containing uridine with or without 5-FOA at 30 °C for 48 h; then cells were harvested and processed for NGS analysis.

In the screen for fluconazole-resistant mutants, transposition-mediated mutagenesis was performed using strain YW02 grown in YPD plate containing Dox for 24 h at 30 °C. Then, the cells were replica-transferred to GMM plates and incubated at 30 °C for 24 h to enrich Arg+Ura+ cells that had undergone the transposition. Then, mutant cells were collected and washed three times with PBS. Approximately $2–4 \times 10^6$ cells/mL were cultivated further in 20 mL YPD medium in the presence or absence of fluconazole. Cell growth was monitored by measuring $OD_{600}$, and cells were harvested after the population had doubled 10 times and processed for NGS analysis.

**Fluconazole susceptibility test**. Antifungal susceptibility testing of the strains included in this study was performed by using the broth microdilution method described in the CLSI document M27-A3[67] in a 96-well plate format.

Fluconazole concentrations tested ranged from 64 to 0.25 µg/mL. RPMI 1640 medium (Thermo Fisher Scientific, 21875109) was prepared according to the CLSI document. The medium was buffered with 0.165 M morpholinepropanesulfonic acid (MOPS), and its pH was adjusted using NaOH and HCl. Cell inoculum was $\sim 1 \times 10^3$ cells per well. Plates were incubated without shaking at 30 °C for 24 h. The content of each well was carefully resuspended by pipetting up and down before $OD_{600}$ was measured using the Spark multimode microplate reader (TECAN, Switzerland).

**Preparation of *C. albicans* plasma membrane**. Highly purified plasma membrane was isolated as previously described for *S. cerevisiae* with minor modifications. *C. albicans* yeast cells were harvested by centrifugation (5000*g*, 5 min), then washed twice with 0.4 M sucrose (Sigma, V900116-500G) in buffer A (25 mM imidazole, pH 7.0 adjusted by HCl). Enough 0.4 M sucrose in buffer A and glass beads were added to the cell pellet and the mix was vortexed for 1 min 8–10 times, with 2 min on ice between rounds. Then the mix was centrifuged at 530*g* (2500 rpm in Sorvall SS34 rotor) for 20 min to pellet unbroken cells and glass beads, and the supernatant was centrifuged at 22,000*g* (16,000 rpm in Sorvall SS34 rotor) for 30 min to obtain a pellet that includes the plasma membranes and mitochondria. The pellet was resuspended in buffer A by gentle vortexing for 30 s and the resuspended membranes were loaded onto discontinuous sucrose gradients (overlaying three 4 mL layers of 2.25, 1.65, and 1.1 M sucrose in buffer A in a $14 \times 89$ mm Beckman ultraclear tube) and centrifuged either overnight (14 h) at 80,000*g* (22,000 rpm) or for 6 h at 284,000*g* (40,000 rpm) in the Beckmann SW41 or SW40Ti rotor. The membrane band at the 2.25/1.65 M sucrose interface is essentially pure plasma membrane, which was collected at these interfaces from the top of the gradient with a Pasteur pipette. The collected membrane was diluted four times with buffer A and pelleted at 30,000*g* (18,000 rpm Beckman 50 Ti rotor) for 40 min. The plasma membrane pellet was suspended in 80% methanol and stored at −80 °C for sterol analysis.

**Sterol analysis**. Sterols were extracted using pre-chilled 80% (v/v) methanol (Sigma, 34860-1L-R) according to the method of Yuan et al.[68]. *C. albicans* strains were cultured in YPD medium with or without fluconazole for 24 h, and the PBS washed cells were harvested by centrifugation. Add pre-chilled 80% (v/v) methanol and glass beads to the cell pellets. Vortex 1 min, then keep on ice for 2 min. Repeat this step 8 or 10 more times. Centrifuge at 14,000*g* for 40 min using a refrigerated centrifuge at 4 °C to obtain the supernatant. The extracts were vacuum-dried and stored at −80 °C. The dried samples were solubilized in methanol prior to analysis by LC-MS. Equal amounts of cells were processed for protein extraction, and protein concentration was measured by Pierce™ BCA Protein Assay Kit (Thermo Fisher Scientific, 23227) to conduct normalization.

LC-MS analysis was performed with a UPLC system, which was coupled to a Q-Exactive orbitrap mass spectrometer (Thermo Fisher, CA) equipped with an atmospheric pressure chemical ionization (APCI) probe. Sterol extracts were separated by a Kinetex® $100 \times 2.1$ mm 2.6 µm column (Phenomenex). A binary solvent system was used, in which mobile phase A consisted of 100% $H_2O$ (0.1% FA), 10 mM ammonium acetate, and mobile phase B of 100% ACN (0.1% FA). A 10-min gradient with a flow rate of 300 µL/min was used. Column chamber and sample tray were held at 45 and 10 °C, respectively. Data with mass ranges of *m/z* 70–150 was acquired at a positive ion mode with data dependent MSMS acquisition. The full scan and fragment spectra were collected with a resolution of 70,000 and 17,500, respectively. The source parameters are as follows: discharge current: 6 µA; capillary temperature: 350 °C; heater temperature: 300 °C; sheath gas

flow rate: 35 Arb; auxiliary gas flow rate: 15 Arb. Data analysis was performed by the software Xcalibur (Thermo Fisher, CA). Compounds were identified based on their retention times and the masses detected in the liquid chromatography-mass spectrometry system.

**Lipid analysis**. Lipids were extracted according to the method of Bligh and Dyer[69]. *C. albicans* strains were cultured in YPD medium with or without fluconazole for 24 h, and the PBS washed cells were harvested by centrifugation. The cell pellets were lysed in PBS by bead-beating mechanical disruption at 4 °C. The supernatants were then extracted with chloroform (AMRESCO, 0757-500ML)/methanol (2:1) at a final ratio of 20% (v/v). Vortex 30 s, then keep for 2 min. Repeat this step 5 more times. Centrifuge at 3000*g* for 20 min using a refrigerated centrifuge at 4 °C to obtain the supernatant. The extracts were evaporated to dryness under $N_2$ at room temperature and stored at −80 °C. The dried samples were solubilized in dichloromethane (Sigma, 650463-1L) methanol (2:1) before analysis by LC-MS. Equal amounts of cells were processed for protein extraction and protein concentration was measured by BCA assay to conduct normalization.

LC-MS analysis was performed with a UPLC system was coupled to a Q-Exactive orbitrap mass spectrometer (Thermo Fisher, CA) equipped with a heated electrospray ionization (HESI) probe. Lipid extracts were separated by a CORTECS C18 $100 \times 2.1$ mm 1.9 µm column (Waters). A binary solvent system was used, in which mobile phase A consisted of ACN:$H_2O$ (60:40), 10 mM ammonium acetate, and mobile phase B of IPA:ACN (90:10), 10 mM ammonium acetate. A 35-min gradient with a flow rate of 220 µL/min was used. Column chamber and sample tray were held at 45 and 10 °C, respectively. Data with mass ranges of *m/z* 240–2000 and *m/z* 200–2000 was acquired at positive ion mode and negative ion mode with data dependent MSMS acquisition. The full scan and fragment spectra were collected with a resolution of 70,000 and 17,500, respectively. The source parameters are as follows: spray voltage: 3000 V; capillary temperature: 320 °C; heater temperature: 300 °C; sheath gas flow rate: 35 Arb; auxiliary gas flow rate: 10 Arb. Data analysis and lipid identification were performed by the software Lipidsearch 4.0 (Thermo Fisher, CA). Compounds were identified based on their retention times and the masses detected in the liquid chromatography-mass spectrometry system.

## Data availability

The sequencing data were deposited to NCBI under BioProject ID PRJNA486232. The authors declare that all other data supporting the findings of this study are available within the article and its Supplementary Information files, or from the corresponding authors upon request.

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

## Acknowledgements

We thank Dr. Yang Chen from Dr. Li Yu's lab and Dr. Feng Chen for technical assistance. We are grateful for the support of Metabolomics Facility in Technology Center for Protein Sciences and the advice of members of J.W.'s group and Yanyi Huang's group at various stages of this work. J.G. thanks Jing Li for her support. This work was supported by the Thousand Young Talents Program (J.W.), Ministry of Science and Technology of China (2016YFC0900103 to J.W.), National Natural Science Foundation of China (21675098 to J.W.), THU-PKU Center for Life Sciences (J.G. and J.W.), and the Agency for Sciences, Technology and Research of Singapore (BMRC/BnB/0001b/2012 to Y.W.). Funding includes NIH GM117111 to H.L. We are grateful to Dr. Li-lin Du of NIBS for providing pPB[ura4] and pDUAL-PBase and Allan Bradley of Sanger for hyPBase.

## Author contributions

J.G., J.W., and Y.W. conceptualized, designed, and supervised the project; H.L. conceptualized, designed and supervised the project and acquired funding; J.G. conducted experiments; H.W. and Z.L. contributed to the computational analysis; A.H.-H.W., Y.-Z. W., Y.G., and X.L. provided technical assistance; G.Z. generated haploid strains; J.G., H. W., J.W., and Y.W. analyzed data and wrote the paper; J.W. and Y.W. acquired funding.

## Additional information

**Competing interests:** The authors declare no competing interests.

