## [Peer Review File · Nature Communications]

Reviewers' comments:

Reviewer #1 (Remarks to the Author):

The authors report the results of their development of the PiggyBac Transposon system for use in *Candida albicans*. The studies are thoughtfully conceived, completed, and interpreted. The authors use complementary lines of genetic, phenotypic, and biochemical evidence and appropriate controls to demonstrate two significant observations. First, they show the technology they have adapted for whole genome analysis. Second, they identify a new component of the triazole mechanism of action and resistance mechanism. I do not have recommendations to improve the manuscript beyond the limitations the authors themselves note.

Reviewer #2 (Remarks to the Author):

In this manuscript, Gao et al. describe a novel strategy for functional genomic analysis in the human pathogen *Candida albicans*, using a transposon mutagenesis strategy in haploid strains. They describe and utilize this technology to identify previously uncharacterized factors involved in resistance to the azole antifungal fluconazole - and in particular, follow up on Fen1 and Fen12, involved in sphingolipid biosynthesis. The authors further perform follow up characterization of these factors, and the connection between sphingolipid biosynthesis and azole resistance. The strength of this work is the introduction of a powerful new technology for functional genetics in an important fungal pathogen, and an example of how this tool can be effectively exploited. It is also well-written and clear. While the technological advancement presented will be of particular interest, some of the follow-up analysis and interpretation requires some revisions to further improve the impact of the manuscript. Overall, the specific proposed mechanism of Fen-mediated resistance to azoles needs to be refined.

Specific points:

1. In the introduction, the authors emphasize how limited genome-wide screens have been in *C. albicans*. While it is true that there have been many limitations to working in this pathogen vs. *S. cerevisiae*, the authors overlook (and fail to cite) some major works of significance to functional genomics in *Candida*. Examples: Roemer et al 2003, Homann et al. 2009, Noble et al 2010, Ryan et al 2012, O'Meara et al 2015, Lee et al 2016 ...
2. The authors should additionally reference the recent paper (Mielich et al 2018. G3), which describes a transposon based mutagenesis system for *C. albicans* - and perhaps emphasize the differences or relative advantages to their system.
3. For Figure 1B: what is the explanation for any colony growth in the absence of Doxycycline? If PBase is required to excise and integrate - why are there any transformations in the absence of Dox? Are they spontaneous URA mutants? if so, will this be confounding with the actual transformations? In addition - the authors could include quantification of these results rather than one plate image.
4. In the results section focusing on genome-wide distribution of transposons, the authors state that inability to generate mutations in certain genes is likely due to their essentiality. Since the nature of essential genes is of considerable interest, it would be worth including a table that lists these putative essential genes, and also commenting on a comparison between these genes and those characterized as essential from other screens (ie Roemer et al 2003).
5. The section on identifying URA mutations using the transposon system would do better to be shortened in text and moved to supplementary, since this does not reveal new biology, or much new insight into the technology developed in this manuscript. The level of detail in this section detracts from the rest of the text. Additionally, it is unclear why the authors have chosen to perform NGS analysis on the mutant pools in order to determine the genes mutated in four 5-FOA

resistant clones (first paragraph, page 14). Why would the inverse PCR not simply be performed on these 4 isolates?

6. The authors use Dox to induce the transposition system, and treat with fluconazole to identify drug resistance mutations. Since previous work has shown synergistic effects between dox and fluconazole - could this impact the results? This should be discussed.

7. Throughout the text, the authors use an agar spotting assay to monitor fluconazole resistance. Since this is a major phenotype being addressed in this manuscript, and one that is well studied in *C. albicans*, the authors should (a) quantify this phenotype by CFU counts, and (b) use microbroth dilutions to calculate an MIC value, since this is the clinical standard for antifungal drug resistance. In some spotting figures, the subtle changes in resistance are difficult to interpret by eye, so this will help quantify these results.

8. Figure 5 indicates a change in ergosterol levels as well as toxic sterol accumulation in wt vs. fen mutant cells. Compared to *erg3* mutants, this is a very small change. Can the authors provide evidence from previous reports, as to whether such small changes in toxic sterol accumulation are known/are sufficient to impact azole resistance?

9. In Figure 5B, these numbers could be depicted in a heat map, so the readers can more readily understand which fatty acids etc are increased/decreased in different mutant strains?

10. Previous work (Sharma et al 2014 AAC) has shown the role of Fen1/Fen12 in resistance to amphotericin B. This work finds that these mutants are hypersensitive to ampB, compared to the azole resistance observed in this manuscript. It would be relevant to cite this work, and reconcile in text this observed phenotype of ampB sensitivity, to azole resistance, given that both classes of drugs target ergosterol.

11. The results and their interpretations in Figure 7a are problematic. The authors state that deleting one copy of LAG1 in *fen12* mutant or one copy of UPC2 in *fen1* mutant reduces resistance. Firstly, some of the spottings are difficult to interpret (see point 7). Secondly, deletion of the *fen* genes does not appear to have any additional resistance phenotype compared to *lag1* or *upc2* heterozygous mutations. For instance, the *upc2* heterozygous deletion is sensitive to fluconazole compared to WT and the *fen1* homozygous deletion is more resistant. Stating that deleting one copy of UPC2 in the *fen1* mutant reduces resistance is a misleading statement, seeing as how the *upc2* het strain already has reduced resistance. The authors should either: omit this analysis, reinterpret the data, or use quantitative resistance data (ie point 7) to calculate genetic interactions using accepted methods (ie Baryshnikova et al 2010 Nat Methods). Same goes for *upc1/fen12* double homozygous mutant.

12. Similar to point 9 - UPC2 and FEN mutant data should be included interrupted together for figure 9B. In this case, mRNA data of *fen* mutants on their own is presented in figure 8, while *fen/upc* double mutants is presented alongside *upc* single mutants in figure 9b. In order to properly interpret interactions between these factors, this should be displayed together on a single graph.

Minor points:

1. The authors refer to *C. albicans*' 'obligate diploidy' in the abstract - however this should be rephrased given the authors are working with haploid strains.

2. The sentence on page 2, line 60 "An alarming trend...", should include a citation.

3. I find it confusing when manuscripts refer to specific strains using a unique lab identification system (ie GZY803, YW01). Even though these strains are mostly described in the text, I think it would be beneficial to call them by a more generic and informative name throughout the text.

4. Similarly, GMM is not a 'standard' media - could the authors refrain from using this term in the main text, and refer to 'synthetic defined' or a more commonly understood media?

5. In the last paragraph on page 17, sometimes the authors use fold change, and other time percentage change when describing similar effects - this should be made consistent to better allow readers to compare differences.

6. It would be helpful if heterozygous strains (ie figure 7) were written as *UPC2/upc2Δ* instead of *upc2Δ*, since it is confusing with haploid genotypes.

Reviewer #3 (Remarks to the Author):

The manuscript by Gao et al describes a creative and innovative method to transposon mutagenize *Candida albicans* using an internally expressed transposon, a regulatable transposase, and haploid *C. albicans* strains. This method is very easy and as the authors point out can be done simply by growing cells on media containing doxycycline and arg- and ura- conditions. They convincingly show that it is very easy and effective. They use it to look for fluconazole resistant mutants and find *fen1* Δ and *fen12* Δ . These mutations are shown to have an impact on fluconazole resistance in diploid strain BWP17 as well. They find that these mutations cause decreases in long chain fatty acids in some ceramide and IPC and MIPC species, as well as increases in shorter chain forms of these molecules. This also decreases plasma membrane distribution of DCMDD levels, which may help alleviate toxicity. Finally, they find that treatment with fluconazole itself also increases production of ceramides and downstream products, and overexpression of IPT1 increases fluconazole resistance.

The method introduced here is exciting and has great potential. It has the drawback that haploid *C. albicans* are not virulent, but the approach can still be used as they demonstrate to find genes influencing metabolic processes and then move this into diploids. The results with *fen1* Δ and *fen12* Δ are interesting, but not particularly ground breaking given that a relationship is already known between sphingolipid synthesis and drug resistance. Given that mutations in sphingolipid biosynthesis can negatively impact virulence, it is not clear to me that *Fen1* or *Fen12* could be involved in drug resistance in animals due to fitness costs. However the role that sphingolipid upregulation plays during fluconazole is potentially important. With this in mind, they did little to probe the mechanism by which changes in sphingolipid levels were impacting drug resistance. For example, it has been implicated from other work (Prasad et al *Antimicrob Agents Chemother.* 2005 Aug; 49(8): 3442-52) that IPT1 impacts drug efflux via *Cdr1*. Furthermore, they suggest that one mechanism might be that sphingolipids impact DCMDD localization, but this was not explored further.

Response to reviewers' comments:

Reviewer #1 (Remarks to the Author):

The authors report the results of their development of the PiggyBac Transposon system for use in *Candida albicans*. The studies are thoughtfully conceived, completed, and interpreted. The authors use complementary lines of genetic, phenotypic, and biochemical evidence and appropriate controls to demonstrate two significant observations. First, they show the technology they have adapted for whole genome analysis. Second, they identify a new component of the triazole mechanism of action and resistance mechanism. I do not have recommendations to improve the manuscript beyond the limitations the authors themselves note.

We thank the reviewer for his full support of this work.

Reviewer #2 (Remarks to the Author):

In this manuscript, Gao et al. describe a novel strategy for functional genomic analysis in the human pathogen *Candida albicans*, using a transposon mutagenesis strategy in haploid strains. They describe and utilize this technology to identify previously uncharacterized factors involved in resistance the azole antifungal fluconazole - and in particular, follow up on Fen1 and Fen12, involved in sphingolipid biosynthesis. The authors further perform follow up characterization of these factors, and the connection between sphingolipid biosynthesis and azole resistance. The strength of this work is the introduction of a powerful new technology for functional

genetics in an important fungal pathogen, and an example of how this tool can be effectively exploited. It is also well-written and clear. While the technological advancement presented will be of particular interest, some of the follow-up analysis and interpretation requires some revisions to further improve the impact of the manuscript. Overall, the specific proposed mechanism of Fen-mediated resistance to azoles needs to be refined.

Specific points:

2.1. In the introduction, the authors emphasize how limited genome-wide screens have been in *C. albicans*. While it is true that there have been many limitations to working in this pathogen vs. *S. cerevisiae*, the authors overlook (and fail to cite) some major works of significance to functional genomics in *Candida*. Examples: Roemer et al 2003, Homann et al. 2009, Noble et al 2010, Ryan et al 2012, O'Meara et al 2015, Lee et al 2016 ...

Apologies for overlooking some previous works. We have cited these references in the revised manuscript. Please see references 6-10.

2.2 The authors should additionally reference the recent paper (Mielich et al 2018. G3), which describes a transposon based mutagenesis system for *C. albicans* - and perhaps emphasize the differences or relative advantages to their system.

As suggested, we have mentioned this work (Please see lines 351-353 and reference 42). In fact, some of us are coauthors of the paper.

2.3. For Figure 1B: what is the explanation for any colony growth in the absence of Doxycycline? If PBase is required to excise and integrate - why are there any transformations in the absence of Dox? Are they spontaneous URA mutants? if so, will this be confounding with the actual transformations? In addition - the authors could include quantification of these results rather than one plate image.

In the absence of Dox (Fig. 1b), there is basal expression of the transposase from the Tet-On promoter which catalyzes a low level of transposition, thus the formation of a small number of colonies on GMM plates. We spread $\sim 2 \times 10^7$ cells that were grown in Dox-free medium onto GMM plates and obtained 40-50 colonies on each, indicating that the basal level of transposition occurred in $<0.001\%$ of the cells. In the text, we have added a sentence to explain the result (please see lines 118-119).

2.4. In the results section focusing on genome-wide distribution of transposons, the authors state that inability to generate mutations in certain genes is likely due to their essentiality. Since the nature of essential genes is of considerable interest, it would be worth including a table that lists these putative essential genes, and also commenting on a comparison between these genes and those characterized as essential from other

screens (ie Roemer et al 2003).

We understand the interest of this reviewer in the essential genes suggested by our NGS analysis of the mutant library. In fact, we are collaborating with Judith Berman's group at Tel Aviv University to conduct genome-wide screens of essential genes in *C. albicans* using two different transposon mutagenesis systems. As the essentiality of many genes is conditional, results obtained in different studies under different conditions need careful comparison, and such comparisons require significant efforts. Recent publications by Rancati's group significantly changed the definition of gene essentiality (reference 1 and 2). Therefore, we believe that the list of essential genes suggested by our screen is not ready for publication, which we hope the reviewer can kindly agree. We also believe that leaving out the list of potentially essential genes will not in any way weaken either the quality or the impact of this study in spite that including it may interest more readers.

(1) Rancati et al. (2018) Emerging and evolving concepts in gene essentiality. *Nat Rev Genet* 19:34.

(2) Liu et al. (2017) Gene Essentiality Is a Quantitative Property Linked to Cellular Evolvability. *Cell* 163:1388

2.5. The section on identifying URA mutations using the transposon system would do better to be shortened in text and moved to supplementary, since this does not reveal new biology, or much new insight into the technology developed in this manuscript. The level of detail in this section detracts from the rest of the text. Additionally, it is unclear why the authors have chosen to perform NGS analysis on the mutant pools in order to determine the genes mutated in four 5-FOA resistant clones (first paragraph, page 14). Why would the inverse PCR not simply be performed on these 4 isolates?

We beg to differ with the reviewer on this issue. First, as we are developing a new mutagenesis system in *C. albicans*, we believe that it is very important to show how well the system works in quantitative terms. Screening for and characterization of 5-FOA resistant mutants, whose mechanism of resistance is well understood, served this purpose well. The successful identification of *PB* insertion in *URA3* or *URA5* in all 26 true 5-FOA mutants by inverse PCR is a strong demonstration of the efficiency and power of this mutagenesis system. Second, as a screen often yields a large number of cells or colonies, it is very time-consuming and labor-intensive to map the *PB* insertion site in each colony by inverse PCR, while NGS allows rapid analysis of the entire 5-FOA-resistant mutant pool and can reveal the majority, if not all, of the genes with a *PB* insert. In this study, we showed that inverse PCR and NGS both identified *URA3* and *URA5*. Thus, we believe that it is important to keep this part in the main text. However, respecting the comments of the reviewer, we have revised and shortened this part as best as we could.

2.6. The authors use Dox to induce the transposition system, and treat with fluconazole to identify drug resistance mutations. Since previous work has shown

synergistic effects between dox and fluconazole - could this impact the results? This should be discussed.

We apologize that the previous description caused some confusion. Dox and fluconazole were not used together. Dox was only used to generate the mutant pool and was not present in the medium during the subsequent screening process such as fluconazole treatment. This was made clear in the text. Please see lines 234-236.

2.7. Throughout the text, the authors use an agar spotting assay to monitor fluconazole resistance. Since this is a major phenotype being addressed in this manuscript, and one that is well studied in *C. albicans*, the authors should (a) quantify this phenotype by CFU counts, and (b) use microbroth dilutions to calculate an MIC value, since this is the clinical standard for antifungal drug resistance. In some spotting figures, the subtle changes in resistance are difficult to interpret by eye, so this will help quantify these results.

We thank the reviewer for the suggestion. We have replaced the results of the spot assays with those of broth microdilution assays in Figure 4b, 4e and 6c.

2.8. Figure 5 indicates a change in ergosterol levels as well as toxic sterol accumulation in wt vs. fen mutant cells. Compared to *erg3* mutants, this is a very small change. Can the authors provide evidence from previous reports, as to whether such small changes in toxic sterol accumulation are known/are sufficient to impact azole resistance?

This is an excellent point. We are not aware of any previous studies correlating the level of toxic sterol molecules in the plasma membrane with azole resistance. Although in comparison with *urg3* mutants, the level of the toxic sterol in the plasma membrane of *fenΔ/Δ* cells is high, it still represents a ~70% decrease compared with the wild type, which, we believe, will at least contribute to the observed resistance. We discussed in great details on how the increase of sphingolipids, as a result of deleting *FEN1* or *FEN12*, in the cell membrane may increase the cell's tolerance of the toxic sterol. Please see lines 422-438 of Discussion.

2.9. In Figure 5B, these numbers could be depicted in a heat map, so the readers can more readily understand which fatty acids etc are increased/decreased in different mutant strains?

As suggested we have changed the data to a heatmap. Please see Figure 5b.

2.10. Previous work (Sharma et al 2014 AAC) has shown the role of Fen1/Fen12 in resistance to amphotericin B. This work finds that these mutants are hypersensitive to ampB, compared to the azole resistance observed in this manuscript. It would be relevant to cite this work, and reconcile in text this observed phenotype of ampB

sensitivity, to azole resistance, given that both classes of drugs target ergosterol.

We thank the reviewer for reminding us the paper by Sharma et al. The increased sensitivity of *fen* mutants to amphotericin B (AmB) observed by Sharma and colleagues can be explained by the different mode of action of this antifungal drug and the way it interacts with ergosterol and sphingolipids. It is widely accepted that amphotericin B binds to ergosterol in a parallel manner to form barrel-stave type pores and penetrate cell membranes, in which their hydrophilic polyhydroxy side pointing inward to constitute the pore lining and their hydrophobic lipophilic heptaene part directing outward to interact with the membrane interior. While the high levels of sphingolipids in the membrane hinder the insertion of the hydrophilic toxic sterol, the binding of amphotericin B to ergosterol is not affected; and on the contrary, the shorter sphingolipids in *fen* Δ/Δ mutants may even expose ergosterol molecules and thus facilitate the binding to AmB, making mutant cells more sensitive to amphotericin B.

We have cited the work by Sharma et al. (reference 62) and discussed their observation. Please see line 438-447.

2.11. The results and their interpretations in Figure 7a are problematic. The authors state that deleting one copy of LAG1 in *fen12* mutant or one copy of UPC2 in *fen1* mutant reduces resistance. Firstly, some of the spottings are difficult to interpret (see point 7). Secondly, deletion of the *fen* genes does not appear to have any additional resistance phenotype compared to *lag1* or *upc2* heterozygous mutations. For instance, the *upc2* heterozygous deletion is sensitive to fluconazole compared to WT and the *fen1* homozygous deletion is more resistant. Stating that deleting one copy of UPC2 in the *fen1* mutant reduces resistance is a misleading statement, seeing as how the *upc2* het strain already has reduced resistance. The authors should either: omit this analysis, reinterpret the data, or use quantitative resistance data (ie point 7) to calculate genetic interactions using accepted methods (ie Baryshnikova et al 2010 Nat Methods). Same goes for *upc2/fen12* double homozygous mutant.

We have replaced the data with the result of broth microdilutions and rewritten the description. Please see the paragraph starting with line 326 and Figure 6c.

2.12. Similar to point 9 - UPC2 and FEN mutant data should be included interrupted together for figure 7B. In this case, mRNA data of *fen* mutants on their own is presented in figure 6, while *fen/upc* double mutants is presented alongside *upc* single mutants in figure 7b. In order to properly interpret interactions between these factors, this should be displayed together on a single graph.

We have merged the two parts by repeating the qPCR analysis. Please see Figure 6b.

Minor points:

2.13. The authors refer to *C. albicans*' 'obligate diploidy' in the abstract - however this should be rephrased given the authors are working with haploid strains.

We removed the word 'obligate'. As haploid *C. albicans* strains have been isolated only under some laboratory conditions that forced the cells to undergo concerted chromosome loss, the haploid strains found were rare products that happened to have lost a complete set of chromosome. So far, no clinical isolates have been found to be haploid. Thus, the isolation of some haploid strains has not fundamentally changed the concept that *C. albicans* is a diploid organism although they are very useful for research.

2.14. The sentence on page 2, line 60 "An alarming trend...", should include a citation.

We have added a reference.

2.15. I find it confusing when manuscripts refer to specific strains using a unique lab identification system (ie GZY803, YW01). Even though these strains are mostly described in the text, I think it would be beneficial to call them by a more generic and informative name throughout the text.

In research papers, the use of own lab identifiers of bacterial or fungal strains is very common. We have been using lab identifiers in dozens of our previous publications in various journals. As the strains were generated in our labs and their relevant traits or genotypes were clearly defined when first mentioned in the text, we do not expect the readers to have problems.

2.16. Similarly, GMM is not a 'standard' media - could the authors refrain from using this term in the main text, and refer to 'synthetic defined' or a more commonly understood media?

We explained that GMM means glucose minimal medium when it first appeared in the text, and the recipe is given in Methods.

2.17. In the last paragraph on page 17, sometimes the authors use fold change, and other time percentage change when describing similar effects-this should be made consistent to better allow readers to compare differences.

We have revised the description to be consistent.

2.18. It would be helpful if heterozygous strains (ie figure 7) were written as UPC2/upc2 Δ instead of upc2 Δ , since it is confusing with haploid genotypes.

We have modified the strain names as suggested.

Reviewer #3 (Remarks to the Author):

The manuscript by Gao et al describes a creative and innovative method to transposon mutagenize *Candida albicans* using an internally expressed transposon, a regulatable transposase, and haploid *C. albicans* strains. This method is very easy and as the authors point out can be done simply by growing cells on media containing doxycycline and arg- and ura- conditions. They convincingly show that it is very easy and effective. They use it to look for fluconazole resistant mutants and find *fen1*Δ and *fen12*Δ. These mutations are shown to have an impact on fluconazole resistance in diploid strain BWP17 as well. They find that these mutations cause decreases in long chain fatty acids in some ceramide and IPC and MIPC species, as well as increases in shorter chain forms of these molecules. This also decreases plasma membrane distribution of DCMDDD levels, which may help alleviate toxicity. Finally, they find that treatment with fluconazole itself also increases production of ceramides and downstream products, and overexpression of IPT1 increases fluconazole resistance.

The method introduced here is exciting and has great potential. It has the drawback that haploid *C. albicans* are not virulent, but the approach can still be used as they demonstrate to find genes influencing metabolic processes and then move this into diploids. The results with *fen1*Δ and *fen12*Δ are interesting, but not particularly ground breaking given that a relationship is already known between sphingolipid synthesis and drug resistance. (3.1) Given that mutations in sphingolipid biosynthesis can negatively impact virulence, it is not clear to me that *Fen1* or *Fen12* could be involved in drug resistance in animals due to fitness costs.

We appreciate the concern of the reviewer regarding the possible fitness cost associated with the deletion of *FEN1* and *FEN12*. We are fully aware that mutations that cause resistance to a drug often incur a fitness cost on the pathogen. However, it is also known that drug-resistant mutants, in both bacteria and fungi, can further evolve to improve fitness through accumulating compensatory mutations (see references 1-4 below). The resistance allows the pathogen to survive during drug treatment and thus gain opportunities to evolve. We certainly plan to evaluate the fitness of the *FEN* mutants both in the presence and absence of fluconazole and carry out evolution studies in future but it is beyond the scope of this study.

References

- (1) Cowen et al. (2001) Divergence in Fitness and Evolution of Drug Resistance in Experimental Populations of *Candida albicans*. *J Bac* 183, 2971.
- (2) Sasse et al. (2012) The stepwise acquisition of fluconazole resistance mutations causes a gradual loss of fitness in *Candida albicans*. *Mol Microbiol* 86, 539.
- (3) Popp et al. (2017) Competitive Fitness of Fluconazole-Resistant Clinical *Candida albicans*. *Antimicrobial Agents and Chemotherapy* 61, 1.
- (4) Hill et al. (2015) Fitness Trade-Off Associated with the Evolution of Resistance to Antifungal Drug Combinations. *Cell Rep* 10, 809.

However the role that sphingolipid upregulation plays during fluconazole is potentially important. With this in mind, they did little to probe the mechanism by which changes in sphingolipid levels were impacting drug resistance. For example, it has been implicated from other work (Prasad et al Antimicrob Agents Chemother. 2005 Aug;49(8):3442-52) that IPT1 impacts drug efflux via Cdr1. (3.2) Furthermore, they suggest that one mechanism might be that sphingolipids impact DCMDD localization, but this was not explored further.

The work by Prasad showed that *IPT1* depletion led to the mislocalization of drug pump Cdr1 under drug-free conditions. A deduction could be that *IPT1* overexpression leads to drug efflux by Cdr1. However, accumulation of toxic sterol shown in Fig 5a does not support significant drug efflux.

Based on physical chemistry features of the toxic sterol and sphingolipids, we proposed that increased sphingolipids in the membrane of the *fen* mutants hindered the insertion of the toxic sterol into the plasma membrane. We have discussed in great details to elucidate our model based on our data and previous works by other scientists. Please see lines 422-438 of Discussion.

REVIEWERS' COMMENTS:

Reviewer #2 (Remarks to the Author):

Gao and colleagues have done an excellent and thorough job of addressing all major points from the initial review. The revised figures and method of displaying certain results add significant clarity and conclusiveness to their findings. I have no additional concerns regarding this manuscript.

Reviewer #3 (Remarks to the Author):

The authors have addressed my comments.